# Building Interactable Replicas of Complex Articulated Objects via Gaussian Splatting

**Yu Liu**[1,2,*,‡]**, Baoxiong Jia**[2,*]**, Ruijie Lu**[2,3]**, Junfeng Ni**[1,2]**, Song-Chun Zhu**[1,2,3]**, Siyuan Huang**[2]
[1]Tsinghua University [2]State Key Laboratory of General Artificial Intelligence, BIGAI [3]Peking University

## Abstract

Building interactable replicas of articulated objects is a key challenge in computer vision. Existing methods often fail to effectively integrate information across different object states, limiting the accuracy of part-mesh reconstruction and part dynamics modeling, particularly for complex multi-part articulated objects. We introduce ArtGS, a novel approach that leverages 3D Gaussians as a flexible and efficient representation to address these issues. Our method incorporates canonical Gaussians with coarse-to-fine initialization and updates for aligning articulated part information across different object states, and employs a skinning-inspired part dynamics modeling module to improve both part-mesh reconstruction and articulation learning. Extensive experiments on both synthetic and real-world datasets, including a new benchmark for complex multi-part objects, demonstrate that ArtGS achieves state-of-the-art performance in joint parameter estimation and part mesh reconstruction. Our approach significantly improves reconstruction quality and efficiency, especially for multi-part articulated objects. Additionally, we provide comprehensive analyses of our design choices, validating the effectiveness of each component to highlight potential areas for future improvement. Our work is made publicly available at: https://articulate-gs.github.io.

## 1 Introduction

Articulated objects, central to everyday human-environment interactions, have become a key focus in computer vision research (Yang et al., 2023a; Weng et al., 2024; Luo et al., 2025; Liu et al., 2024; Deng et al., 2024). Accurately reconstructing real-world scenes (Chen et al., 2024b; Ni et al., 2024; Lu et al., 2024b) and creating interactable digital replicas of these objects are essential for various applications, including scene understanding (Jia et al., 2024; Huang et al., 2024b; Zhu et al., 2024; Linghu et al., 2024) and robotics learning (Liu et al., 2022a; Geng et al., 2023b;a; Gong et al., 2023; Yang et al., 2024b; Zhao et al., 2024; Lu et al., 2024a). By building high-fidelity digital twins of articulated objects, we bridge the gap between synthetic and real-world scenarios, thus facilitating the sim-to-real transfer of robotic systems (Torne et al., 2024; Kerr et al., 2024). As we advance towards more sophisticated robotic systems and immersive virtual environments, there is a growing need for improved and efficient modeling techniques for the reconstruction of articulated objects.

The problem of reconstructing articulated objects has been extensively studied (Liu et al., 2023a;b; Weng et al., 2024; Deng et al., 2024; Yang et al., 2023a), with a key challenge being the learning of object geometry when only partial views of the object are available at any given state. To accurately reconstruct object parts (*e.g.*, a closed drawer), it is essential to integrate observations from multiple object states during interactions (*e.g.*, the opening process of the drawer). This necessitates the simultaneous learning and alignment of fine-grained object parts across different states, which must be achieved jointly during the reconstruction of object geometries. Such a requirement presents significant challenges in object modeling, especially for complex everyday articulated objects that often consist of multiple interactable parts. Additionally, uncertainties in object geometry reconstruction introduce further challenges in modeling articulation, as errors in geometry modeling can result in inaccurate learning of articulation parameters. These challenges highlight the need for improved models that handle the complexities of multi-part articulated objects.

---

*Equal contribution. ‡Work done as an intern at BIGAI.

Recent approaches attempt to address these challenges using part priors from pre-trained models. These models provide either part segmentation masks via models like SAM (Kirillov et al., 2023), or 2D pixel correspondences for aligning pixels across states (Sun et al., 2021). However, these methods rely heavily on priors from pre-trained models, often using single-state inputs and neglecting critical motion information (Mandi et al., 2024) and struggling with the complexity of multi-part objects when accurately matching pixels across states becomes difficult (Weng et al., 2024). These limitations result in unstable and inconsistent learning of object parts, posing significant challenges to the joint learning of part motion and geometry.

To address these challenges, we propose ArtGS, which introduces several key innovations for handling complex multi-part articulated objects. Specifically, we adopt the commonly used two-state setting for learning articulated objects, as established in prior works (Liu et al., 2023a; Weng et al., 2024). Central to our approach is the use of 3D Gaussians (Kerbl et al., 2023) as the foundational representation, chosen for their ability to explicitly maintain spatial information while offering efficiency and high reconstruction quality. To effectively model object dynamics and integrate information across multiple object states, we employ canonical Gaussians with a carefully designed coarse-to-fine initialization and update scheme. These Gaussians act as a bridge between different input object states, enabling accurate deformation modeling that improves both mesh reconstruction and articulation learning. Building on the canonical Gaussians, we draw inspiration from Gaussian skinning (Song et al., 2024) and introduce a center-based clustering module for part and dynamics learning. This approach leverages motion priors of Gaussians, which are summarized during the learning process, serving as a guide to better align object parts between states and improve articulation learning. These designs allow our method to achieve state-of-the-art performance in joint parameter estimation and part mesh reconstruction, excelling on both existing benchmarks and our newly curated complex multi-part articulated object reconstruction benchmark. Our approach outperforms existing methods in both synthetic and real-world scenarios, with significant improvements in axis modeling and overall efficiency. Through extensive experiments, we demonstrate the effectiveness of our model in efficiently delivering high-quality reconstruction of complex multi-part articulated objects. We also provide comprehensive analyses of our design choices, highlighting the critical role of these modules and identifying areas for future improvement.

**Contributions** Our main contributions of this work can be summarized as follows:

- We propose ArtGS, a novel and efficient method for articulated object reconstruction that achieves state-of-the-art performance, particularly for complex multi-part objects.
- We introduce coarse-to-fine canonical Gaussian initialization and skinning-inspired part dynamics modeling with self-guided motion priors to improve object part and articulation learning, effectively addressing the limitations of existing methods in using object motion information.
- We conduct extensive experiments on both synthetic and real-world articulated objects, demonstrating the effectiveness, efficiency, scalability, and robustness of our approach. We also provide comprehensive ablation studies to validate our designs and highlight areas for future improvement.

## 2 RELATED WORK

**Dynamic Gaussian Modeling** Recent advancements have shown the potential of Gaussian Splatting (Kerbl et al., 2023) for 4D reconstruction (Jung et al., 2023; Katsumata et al., 2023; Wu et al., 2024; Luiten et al., 2024; Li et al., 2024; Lu et al., 2024c; Lei et al., 2024; Guo et al., 2024; Qian et al., 2024; Bae et al., 2024; Wan et al., 2024b). A central focus of these efforts is the deformation modeling of 3D Gaussians. While effective for dynamics capturing, most approaches learn transformations implicitly, limiting their capability for controllable dynamics modeling. To address this issue, recent studies use superpoints (Huang et al., 2024c; Wan et al., 2024a) for improved dynamics modeling and control. However, as superpoint learning is based primarily on rendering without considering object physics, these methods fail to reliably capture accurate physical parameters (*e.g.*, joints and axes). Another line of works (Xie et al., 2024; Jiang et al., 2024) introduce controllable Gaussians by integrating physics-based modeling for graphics simulations. These models require intricate priors of objects (*e.g.*, material properties), making them impractical for reconstructing everyday articulated objects. To overcome these challenges, our work combines the explicit 3D Gaussian modeling with articulation modeling, enabling efficient and high-quality reconstruction with precise articulation parameter estimation for more practical digital-twin construction of articulated objects.

**Articulation Parameter Estimation**   Estimating joint articulation parameters for articulated objects has been extensively studied, with approaches broadly categorized into two main categories. First, prediction-based methods estimate joint parameters from sensory inputs of different object configurations (Huang et al., 2014; Katz et al., 2013) or use end-to-end models (Hu et al., 2017; Yi et al., 2018; Li et al., 2020; Wang et al., 2019; Sun et al., 2023; Liu et al., 2022b; Weng et al., 2021; Sturm et al., 2011; Chu et al., 2023; Martín-Martín et al., 2016; Liu et al., 2023c; Gadre et al., 2021; Mo et al., 2021; Jain et al., 2021; Yan et al., 2020; Lei et al., 2023) to predict part segmentation, kinematic structure, as well as joint parameters. Second, reconstruction-based methods optimize articulation parameters by reconstructing multi-view images or videos (Wei et al., 2022; Tseng et al., 2022; Mu et al., 2021; Lewis et al., 2022; Liu et al., 2023a; Lei et al., 2024; Deng et al., 2024; Swaminathan et al., 2024; Noguchi et al., 2022; Zhang et al., 2021; Pillai et al., 2015; Liu et al., 2023b). Most of these methods treat articulation parameter estimation as a separate task, without generating high-quality, interactable part-mesh reconstructions. ArtGS aims to address this gap by integrating part-mesh reconstruction and articulation parameter estimation, enabling the creation of high-quality, interactable replicas.

**Articulated Object Reconstruction**   Articulated object reconstruction, differing from human and animal motion modeling (Joo et al., 2018; Loper et al., 2023; Mihajlovic et al., 2021; Noguchi et al., 2021; Yang et al., 2021b;a; Romero et al., 2022; Zuffi et al., 2017; Yang et al., 2024a; Xu et al., 2020; Tan et al., 2023; Yang et al., 2022; 2023b; Song et al., 2023a; Yang et al., 2023a; Song et al., 2023b), focus on the piece-wise rigidity of each part, requiring both part-level geometry reconstruction and joint articulation parameter estimation. While end-to-end models predict joint parameters and segment object parts from single-stage (Heppert et al., 2023; Wei et al., 2022; Kawana et al., 2021) or interaction observations(Jiang et al., 2022; Ma et al., 2023; Nie et al., 2022; Hsu et al., 2023), they struggle to generalize to unseen objects. Per-object optimization approaches  (Liu et al., 2023a;b; Weng et al., 2024; Deng et al., 2024; Swaminathan et al., 2024), using multi-state observations for articulation modeling, offer better adaptability to unknown objects but face scaling issues of multiple joints. Methods like DTA (Weng et al., 2024) attempt to handle multi-part objects but still struggle with those having more than three movable parts. We address the reliability, flexibility, and scalability issues of previous works with our canonical Gaussian design and skinning-inspired part dynamics modeling, achieving higher accuracy, robustness, and efficiency for articulated object reconstruction.

## 3   PRELIMINARIES

**3D Gaussian Splatting**   3D Gaussian Splatting (3DGS) represents a static 3D scene using 3D Gaussians (Kerbl et al., 2023). Each Gaussian $G_i$ is associated with a center $\boldsymbol{\mu}_i$, covariance matrix $\boldsymbol{\Sigma}_i$, opacity $\sigma_i$ and spherical harmonics coefficients $\boldsymbol{h}_i$. The final opacity of a 3D Gaussian at a spatial point $\boldsymbol{x}$ can be calculated as:

$$\alpha_i(\boldsymbol{x}) = \sigma_i \exp\left(-\frac{1}{2}(\boldsymbol{x} - \boldsymbol{\mu}_i)^T \boldsymbol{\Sigma}_i^{-1}(\boldsymbol{x} - \boldsymbol{\mu}_i)\right), \quad \text{where} \quad \boldsymbol{\Sigma}_i = \boldsymbol{R}_i \boldsymbol{S}_i \boldsymbol{S}_i^T \boldsymbol{R}_i^T. \tag{1}$$

As the physical meaning of a covariance matrix is only valid if it is positive semi-definite, we decompose the covariance matrix $\boldsymbol{\Sigma}_i$ following Eq. (1) into a scaling diagonal matrix $\boldsymbol{S}_i$ and a rotation matrix $\boldsymbol{R}_i$ parameterized by a quaternion $\boldsymbol{r}_i$. A scene is then described with a collection of such Gaussians $\mathcal{G} = \{G_i : \boldsymbol{\mu}_i, \boldsymbol{r}_i, \boldsymbol{s}_i, \sigma_i, \boldsymbol{h}_i\}_{i=1}^N$. We render an image $\boldsymbol{I}$ and optionally its depth image $\boldsymbol{D}$ from the 3D scene $\mathcal{G}$ by projecting each Gaussian onto the 2D image plane and aggregating them using $\alpha$-blending:

$$\boldsymbol{I} = \sum_{i=1}^N T_i \alpha_i^{\text{2D}} \mathcal{SH}(\boldsymbol{h}_i, \boldsymbol{v}_i), \quad \boldsymbol{D} = \sum_{i=1}^N T_i \alpha_i^{\text{2D}} d_i, \quad \text{where} \quad T_i = \prod_{j=1}^{i-1}(1 - \alpha_j^{\text{2D}}). \tag{2}$$

$\alpha_i^{\text{2D}}$ is a 2D version of Eq. (1), with $\boldsymbol{\mu}_i$, $\boldsymbol{\Sigma}_i$, $\boldsymbol{x}$ replaced by the projected $\boldsymbol{\mu}_i^{\text{2D}}$, $\boldsymbol{\Sigma}_i^{\text{2D}}$, and the pixel coordinate $\boldsymbol{u}$. $\mathcal{SH}(\cdot)$ is the spherical harmonic function, $\boldsymbol{v}_i$ is the view direction from the camera to $\boldsymbol{\mu}_i$, $d_i$ is the depth of the $i$-th Gaussian. Given $N_v$ input view images $\{\bar{\boldsymbol{I}}_i, \bar{\boldsymbol{D}}_i\}_{i=1}^{N_v}$, 3DGS learns Gaussians $\mathcal{G}$ with:

$$\mathcal{L}_{\text{render}} = (1 - \lambda_{\text{SSIM}})\mathcal{L}_I + \lambda_{\text{SSIM}}\mathcal{L}_{\text{D-SSIM}} + \mathcal{L}_D, \tag{3}$$

where $\mathcal{L}_I = ||\boldsymbol{I} - \bar{\boldsymbol{I}}||_1$ is the L1-loss, $\mathcal{L}_{\text{D-SSIM}}$ is the D-SSIM loss (Kerbl et al., 2023), $\lambda_{\text{SSIM}}$ is the weight of D-SSIM loss, and $\mathcal{L}_D = \log\left(1 + ||\boldsymbol{D} - \bar{\boldsymbol{D}}||_1\right)$ is the optional depth supervision.

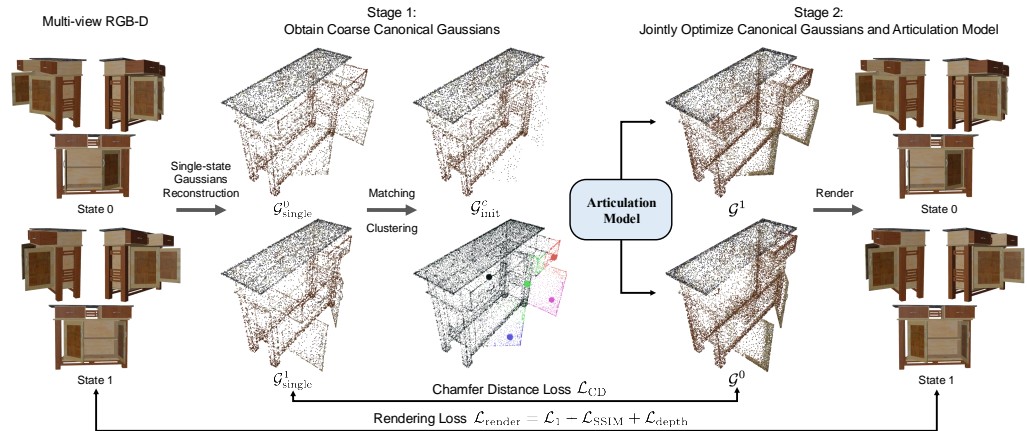

Figure 1: **The overview of ArtGS.** Our method is divided into two stages: (i) obtaining coarse canonical Gaussians $\mathcal{G}^c_{\text{init}}$ by matching the Gaussians $\mathcal{G}^0_{\text{single}}$ and $\mathcal{G}^1_{\text{single}}$ trained with each single-state individually and initializing the part assignment module with clustered centers, (ii) jointly optimizing canonical Gaussians $\mathcal{G}^c$ and articulation model (including the articulation parameters $\Psi$ and the part assignment module in Sec. 4.2).

**Mesh Extraction from Gaussians**  To extract meshes from Gaussian splats $\mathcal{G}$, we can render depth maps and utilize Truncated Signed Distance Function (TSDF) to fuse the reconstructed depth maps, and extract the object mesh $\mathcal{M}$ with marching cubes (Huang et al., 2024a). This process can be done with Open3D (Zhou et al., 2018) with proper choice of voxel size and truncated threshold.

## 4  METHOD

Given $N_v$ RGB-D images of an unknown articulated object $\{\bar{\boldsymbol{I}}^t_i, \bar{\boldsymbol{D}}^t_i\}_{i=1}^{N_v}$ at two joint states $t \in \{0, 1\}$, we aim to reconstruct its part-level meshes $\mathcal{M}$ and joint articulation parameters $\Psi$. We define a set of learnable canonical Gaussians $\mathcal{G}^c$ which can be transformed into joint state Gaussians $\mathcal{G}^t$ via a per-Gaussian SE(3) transformation $T^{c \to t}$, parameterized by $\Psi$. Formally,

$$\mathcal{G}^t = T^{c \to t} \cdot \mathcal{G}_c \quad \text{and} \quad \mathcal{G}^c = (T^{c \to t})^{-1} \cdot \mathcal{G}^t \quad \text{for} \quad t \in \{0, 1\}. \tag{4}$$

We impose the continuity of motion between the joint states by setting the canonical Gaussians $\mathcal{G}^c$ at the mid-state ($c : t = 0.5$), enforcing that $T^{c \to 0} = (T^{c \to 1})^{-1}$. This simplifies the articulation learning and connects the two input joint states through the canonical Gaussians $\mathcal{G}^c$, solving potential issues of occlusion and misinformation when reconstructing object meshes separately on the two joint states.

Using this motion model, we leverage multi-view RGB-D images from the two input states to learn both the canonical Gaussian $\mathcal{G}^c$, the transformation $T^{c \to 1}$ or equivalently the joint parameters $\Psi$, and extract object meshes $\mathcal{M}^t$ for different joint states following Sec. 3. An overview of ArtGS is presented in Fig. 1, with details on key designs provided in the following sections.

### 4.1  COARSE-TO-FINE CANONICAL GAUSSIAN INITIALIZATION WITH MOTION ANALYSIS

The initialization of the canonical Gaussians $\mathcal{G}^c$ is crucial for articulation learning. A good initialization leverages the consistency between input joint states, improving mesh reconstruction and articulation modeling. In contrast, a random initialization leads to undesirable local minima, adversely affecting the learning process (see in Fig. 4). To tackle this issue, we propose a coarse-to-fine strategy for the canonical Gaussian initialization, incorporating preliminary motion information from the two input joint states to enhance subsequent articulation modeling.

**Coarse Initialization by Matching Single-state Gaussians**  In this phase, we first separately train two sets of single-state Gaussians $\mathcal{G}^t_{\text{single}}$ with input multi-view images following Eq. (3). We then apply Hungarian Matching to obtain matched Gaussian pairs between $\mathcal{G}^0_{\text{single}}$ and $\mathcal{G}^1_{\text{single}}$, based on the distance between Gaussian centers. We take the mean of each pair of matched Gaussians as the coarse canonical Gaussian initialization $\mathcal{G}^c_{\text{coarse}}$. To reduce the significant computation time associated

with matching a large number of Gaussians, we use Farthest Point Sampling (FPS) to downsample the learned single-state Gaussians to a set of 5K Gaussians prior to matching.

**Initialization Refinement with Motion Analysis** To support geometry reconstruction and articulation modeling, relying solely on 5K matched coarse Gaussians alone is insufficient. Therefore, we refine the coarse initialization $\mathcal{G}^c_{\text{coarse}}$ guided by the motion information of object parts. Intuitively, single-state Gaussians, $\mathcal{G}^0_{\text{single}}$ and $\mathcal{G}^1_{\text{single}}$, should exhibit consistency for static object parts discrepancies for movable parts, *i.e.*, the static parts of these Gaussians are well-learned. Based on this insight, we refine the set of coarse canonical Gaussians $\mathcal{G}^c_{\text{coarse}}$ by including Gaussians corresponding to static parts, allowing more focused learning of movable parts during articulation modeling. In practice, we classify each Gaussian $G_i$ in a joint state $t$ as static or dynamic by calculating its minimum Chamfer Distance to all Gaussians in the opposite state $\bar{t}$:

$$\text{CD}^{t \to \bar{t}}_i = \min_j ||\boldsymbol{\mu}^t_i - \boldsymbol{\mu}^{\bar{t}}_j||_2, \quad G_i \in \mathcal{G}^t_{\text{single}}, G_j \in \mathcal{G}^{\bar{t}}_{\text{single}} \quad \text{and} \quad \text{CD}^{t \to \bar{t}} = \text{Mean}_i \left( \text{CD}^{t \to \bar{t}}_i \right). \quad (5)$$

If the distance $\text{CD}^{t \to \bar{t}}_i$ exceeds a threshold $\epsilon_{\text{static}}$, $G_i$ is classified as dynamic; otherwise it is static. To determine which state, $t$ or $\bar{t}$, contains more motion information, we compare the mean distance $\text{CD}^{t \to \bar{t}}$ of all Gaussians in state $t$ following Eq. (5) and classify the higher state as the more motion informative state. For instance, a cabinet with open drawers provides clearer identification of movable parts than one with closed drawers. With this information, we add the static Gaussians from the more motion informative state to refine $\mathcal{G}^c_{\text{coarse}}$ into the final initialization of the canonical Gaussian $\mathcal{G}^c_{\text{init}}$.

## 4.2 PART DISCOVERY FOR ARTICULATION MODELING

Following Eq. (4), we use a part-based formulation for articulation modeling. Specifically, given the number of parts $K$, we aim to decompose the Gaussians into $K$ parts and learn the articulation paramerters $\Psi = \{T^{c \to 1}_k\}^K_{k=1}$. In contrast to existing works that leverage prior information for part discovery (Mandi et al., 2024; Weng et al., 2024), we discover parts in an unsupervised manner during learning.

**Center-based Part Modeling and Assignment** Given input canonical Gaussians $\mathcal{G}^c = \{G_i\}^N_{i=1}$, our objective is to compute part-level masks $\mathbf{M} \in \mathbb{R}^{N \times K}$ that assign each Gaussian $\mathcal{G}_i$ to a specific part. A common approach to generating these assignment masks is through unsupervised segmentation modules using MLPs or slot-attention (Locatello et al., 2020; Jia et al., 2023; Liu et al., 2025). However, these models implicitly segment parts and fail to leverage the explicit spatial and dynamic information present in 3D Gaussians. We observe that such methods struggle with parts that exhibit similar motion patterns, leading to incorrect assignments. To address this issue, we adopt a center-based part modeling approach that explicitly utilizes spatial information, inspired by sparse control points from SC-GS (Huang et al., 2024c) and quasi-rigid blend skinning in REACTO (Song et al., 2024). Specifically, we define $K$ learnable centers $C_k = (\boldsymbol{p}_k, \boldsymbol{V}_k, \boldsymbol{\lambda}_k)$ with center location $\boldsymbol{p}_k \in \mathbb{R}^3$, rotation matrix $\boldsymbol{V}_k \in \mathbb{R}^{3 \times 3}$, and scale vector $\boldsymbol{\lambda}_k \in \mathbb{R}^3$. For a given Gaussian $G_i \in \mathcal{G}^c$, we compute the Mahalanobis distance $\boldsymbol{D}_{ik}$ between $G_i$ and center $C_k$ as:

$$\mathbf{X}^k_i = \frac{[\boldsymbol{V}_k(\boldsymbol{\mu}^c_i - \boldsymbol{p}_k)]}{\boldsymbol{\lambda}_k} \quad \mathbf{D}^k_i = (\mathbf{X}^k_i)^T \cdot \mathbf{X}^k_i \quad \text{and} \quad \boldsymbol{M} = \text{GumbelSoftmax} \left( \frac{-\boldsymbol{D} + \boldsymbol{W}_\Delta}{\tau} \right) \quad (6)$$

where $\boldsymbol{D}^k_i$ is the distance matrix for part assignment. One challenge of using the distance matrix for part assignment is identifying sharp boundaries when two parts overlap spatially (*e.g.*, in the case of a closed drawer). To improve boundary identification, we introduce a residual term $\boldsymbol{W}_\Delta = \text{MLP}(\boldsymbol{\mu}, \boldsymbol{X}, \boldsymbol{D})$, predicted by a shallow MLP that concatenates the absolute position of each Gaussian and the distance matrix $\boldsymbol{D}$ as input. This residual is added to the original distance matrix $\boldsymbol{D}$ to refine the part assignment mask following Eq. (6). Notably, we use Gumbel Softmax to ensure that each Gaussian is assigned to only one part, which simplifies the optimization of joint parameters. Detailed implementation can be found in Appendix A.

**Center Initialization by Clustering Coarse Gaussians** We empirically find that the initialization of centers $\boldsymbol{p}_k$ and scale $\boldsymbol{\lambda}_k$ have great impacts on the correctness of part discovery in later learning

process (see Fig. 4). Therefore, similar to the canonical Gaussian initialization described in Sec. 4.1, we utilize the motion type of each joint as additional information for providing good initializations of part centers. Specifically, we select the input joint state with more motion information to identify static and dynamic parts. For static parts, we take the mean of the Gaussians as the part center. For movable parts, we do spectral clustering on the positions of movable Gaussians ($K - 1$ clusters) and take the mean of each cluster for part center initialization. We use the distance from the farthest point to the center of each cluster as the initial scale.

### 4.3 SELF-GUIDED ARTICULATION TYPE AND PARAMETER LEARNING

After obtaining object part representations, we define the per-part articulation parameters via dual-quaternions. Formally, the joints articulation parameters $\Psi = \{T_k^{c \to 1}\}_{k=1}^K = \{q_k^{c \to 1} : (q_{k,r}, q_{k,d})\}_{k=1}^K$, where $q_{k,r}$ and $q_{k,d}$ are the real and dual part of the dual-quaternion that determine the rotation and translation of the joint transformation respectively. For notational simplicity, we use $q_k^t$ for $q_k^{c \to t}$ in the following texts. With the mid-state assumption in Sec. 4, we have $q_k^0 = (q_k^1)^{-1}$ is the inverse of dual-quaternion $q_k^1$. Given object masks $M$ obtained in Sec. 4.2, the per-gaussian dual-quaternion $q_i$ for Gaussian $G_i \in \mathcal{G}^c$ is given by:

$$q_i^t = \left( \sum_{k=1}^K M_{ik} \cdot q_{k,r}^t, \sum_{k=1}^K M_{ik} \cdot q_{k,d}^t \right). \tag{7}$$

where $(q_k^1)^{-1}$ is the inverse of dual-quaternion $q_k^1$ and $M_{ik}$ denotes the probability of Gaussian $i$ belongs to part $k$. With the per-gaussian transformation given $q_i^t$, we transform the canonical Gaussian $\mathcal{G}^c$ to get the two joint state Gaussians $\mathcal{G}^t$ with:

$$\mu_i^t = R_i^{c \to t} \cdot \mu_i^c + t_i^{c \to t}, \quad r_i^t = q_{i,r}^t \otimes r_i^c, \tag{8}$$

where $R_i^{c \to t}$ and $t_i^{c \to t}$ is the per-gaussian rotation matrix and translation vector derived from $q_i^t$, and $\otimes$ denotes quaternion multiplication operation. We assume that the scale $s_i$ and opacity $\sigma_i$ of the Gaussian $G_i$ remains consistent under transformation.

To enhance the learning of articulation parameters, we adopt a warm-up strategy for predicting the joint type of each part. During the warm-up stage, we optimize the articulation parameters $\Psi = \{q_k^1\}_{k=1}^K$ without any constraints. Next, we develop a heuristic for joint type prediction based on the learned rotation $q_{k,r}$. Specifically, we classify the joint as revolute if the rotation degree of $q_{k,r}$ exceeds a threshold $\epsilon_{\text{revol}}$, and otherwise prismatic. With predicted joint types, we constrain the joint transformation for each part. Specifically, we manually set the rotation quaternion $q_{k,r}$ of prismatic joints as identity quaternion. This operation allows the model to focus on optimizing the translation term $q_{k,d}$ of the prismatic joint, thereby obtaining a more accurate estimate of the joint parameters.

### 4.4 OPTIMIZATION

We train our model using the rendering loss with depth supervision $\mathcal{L}_{\text{render}}$ described in Sec. 3 on the reconstructed $\mathcal{G}^t$ for the two joint states as discussed in Sec. 4.3. To reduce the chances of learning artifacts during update, we use the single-state reconstructed Gaussians $\mathcal{G}_{\text{single}}^t$ as an additional supervision:

$$\mathcal{L}_{\text{CD}} = \frac{1}{N} \sum_{i=1}^N \min_j \|\mu_i^t - \mu_j^t\|_2 \quad, \quad G_i \in \mathcal{G}^t, \quad \text{and} \quad G_j \in \mathcal{G}_{\text{single}}^t, \tag{9}$$

where we calculate the single-direction Chamfer Distance between the deformed Gaussians $\mathcal{G}^t$ and single-state reconstructed Gaussians $\mathcal{G}_{\text{single}}^t$ as the loss signal. As these single-state Gaussians are only a rough estimate, we only introduce this loss in the first 1K to 5K steps. Additionally, to regularize the learning of part centers $p_k$, we add another regularization loss as:

$$\mathcal{L}_{\text{reg}} = \frac{1}{K} \sum_{k=1}^K \|p_k - \hat{p}_k\|_2, \quad \text{where} \quad \hat{p}_k = \sum_{i=1}^N \frac{M_{ik}}{\sum_{i=1}^N M_{ik}} \mu_i, \tag{10}$$

which enforces that the centers $p_k$ should be close to the average spatial position of Gaussians in canonical Gaussians $\mathcal{G}^c$ that belong to part $k$. Above all, our supervision could be summarized as:

$$\mathcal{L} = \mathcal{L}_{\text{render}} + \lambda_{\text{CD}} \mathcal{L}_{\text{CD}} + \lambda_{\text{reg}} \mathcal{L}_{\text{reg}}. \tag{11}$$

We provide more implementation and model training details in Appendix A.

## 5 EXPERIMENTS

**Datasets**   We evaluate our method on three datasets: (1) PARIS, a two-part dataset proposed by Liu et al. (2023a), which features articulated objects consisting of one static and one movable part. It includes 10 synthetic objects from the PartNet-Mobility dataset (Xiang et al., 2020) and 2 real-world objects captured using the MultiScan (Mao et al., 2022) toolset. (2) DTA-Multi, a dataset proposed by Weng et al. (2024), containing 2 synthetic multi-part articulated objects from PartNet-Mobility, each with one static part and two movable parts. (3) ArtGS-Multi, our newly curated dataset, featuring 5 complex articulated objects from PartNet-Mobility with 3 to 6 movable parts.

**Metrics**   Following the evaluation protocols of PARIS (Liu et al., 2023a) and DTA (Weng et al., 2024), we assess the performance of all methods using both mesh reconstruction and articulation estimation metrics. For mesh reconstruction, we compute the bi-directional Chamfer Distance between the reconstructed mesh and the ground truth mesh with 10K uniformly sampled points from each mesh. We report the Chamfer Distance for the whole object (CD-w), the static parts (CD-s), and the movable parts (CD-m). For articulation estimation, we evaluate the predicted articulation using the angular error (Axis Ang.) and the distance (Axis Pos., revolute joint only) between the predicted and ground-truth joint axes. We also report the part motion error (Part Motion) which measures the rotation geodesic distance error (in degrees) for revolute joints and Euclidean distance error (in meters) for prismatic joints.

### 5.1 RESULTS ON SIMPLE ARTICULATED OBJECTS

**Experimental Setup**   We use the PARIS dataset as the benchmark and select Ditto (Hsu et al., 2023), PARIS (Liu et al., 2023a), CSG-reg (Weng et al., 2024), 3Dseg-reg (Weng et al., 2024), and DTA (Weng et al., 2024) as baselines for quantitative evaluation. Following the evaluation setting from DTA (Weng et al., 2024), we report all metrics with mean $\pm$ std over 10 trials calculated at the high-visibility joint state. We re-train DTA on the same device (NVIDIA RTX 3090) for training time comparison. Additional results on all joint states are provided in Tab. A.1.

**Results**   As shown in Tab. 1, our method significantly outperforms existing approaches across all metrics, especially for joint articulation parameter estimation, where ArtGS achieves substantially lower errors. This improvement stems from our motion model with Gaussian Splatting, which explicitly deforms Gaussians for more precise part transformation modeling, leading to more precise joint parameter estimation. For mesh reconstruction, ArtGS excels in reconstructing movable parts, yielding lower CD-m values, especially for real-world objects. While DTA performs well on CD-w and CD-s due to its state-by-state reconstruction, we show in Fig. 2 that it struggles with the low-visibility state. In contrast, ArtGS achieves significantly better results on the low-visibility state while maintaining competitive results on the high-visibility state. This is attributed to the canonical Gaussians modeling that connects the two input joint states for mutually improved mesh reconstruction. Additionally, ArtGS shows consistently better results on real-world objects with significantly faster training time, positioning it as an efficient solution for building digital twins of real-world articulated objects.

### 5.2 RESULTS ON COMPLEX ARTICULATED OBJECTS WITH MULTIPLE MOVABLE PARTS

**Experimental Setup**   We use DTA-Multi and ArtGS-Multi as benchmarks for evaluating complex articulated object reconstruction. On DTA-Multi, we compare our model against PARIS and DTA, while on ArtGS-Multi we use DTA as the main baseline given its strong performance. Similar to Sec. 5.1, we report all metrics with a mean over 10 trials for DTA-Multi and 3 trials for ArtGS-multi because of the training time required for baselines. For ArtGS-multi, we report the average of all movable parts for articulation estimation and mesh reconstruction due to the large number of parts. Considering the potential error prediction with no mesh for one of the parts, we manually set the Chamfer Distance of the empty prediction to 1000.

**Results**   As demonstrated in Tab. 2 and Tab. 3, our method consistently outperforms existing methods by a large margin in both mesh reconstruction and articulation estimation. Notably, on ArtGS-Multi, the baseline model DTA struggles with movable part identification and axis prediction

Table 1: **Quantitative evaluation on PARIS.** Metrics are reported as mean ± std over 10 trials at the joint state with higher visibility, following (Weng et al., 2024). PARIS* (Liu et al., 2023a) is augmented with depth for fair comparison. DTA is re-trained for time efficiency comparison. Lower (↓) is better on all metrics and we highlight best and second best results. Objects with † are seen categories trained in Ditto. F indicates wrong motion type predictions. Axis Pos. is omitted for prismatic joints (Blade, Storage, and Real Storage).

| Metric | Method | Synthetic Objects | | | | | | | | | | | Real Objects | | |
| --- | --- | --- | --- | --- | --- | --- | --- | --- | --- | --- | --- | --- | --- | --- | --- |
| | | FoldChair | Fridge | Laptop† | Oven† | Scissor | Stapler | USB | Washer | Blade | Storage† | All | Fridge | Storage | All |
| Axis Ang | Ditto | 89.35 | 89.30 | 3.12 | 0.96 | 4.50 | 89.86 | 89.77 | 89.51 | 79.54 | 6.32 | 54.22 | 1.71 | 5.88 | 3.80 |
| | PARIS* | 15.79±29.3 | 2.93±5.3 | 0.03±0.0 | 7.43±23.4 | 16.62±32.1 | 8.17±15.3 | 0.71±0.8 | 18.40±23.3 | 41.28±31.4 | 0.03±0.0 | 11.14±16.1 | 1.90±0.0 | 30.10±10.4 | 16.00±5.2 |
| | CSG-reg | 0.10±0.0 | 0.27±0.0 | 0.47±0.0 | 0.35±0.1 | 0.28±0.0 | 0.30±0.0 | 11.78±10.5 | 71.93±6.3 | 7.64±5.0 | 2.82±2.5 | 9.60±2.4 | 8.92±0.9 | 69.71±9.6 | 39.31±5.2 |
| | 3Dseg-reg | - | - | 2.34±0.11 | - | - | - | - | - | 9.40±7.5 | - | - | - | - | - |
| | DTA | 0.03±0.0 | 0.09±0.0 | 0.07±0.0 | 0.22±0.1 | 0.10±0.0 | 0.07±0.0 | 0.11±0.0 | 0.36±0.1 | 0.20±0.1 | 0.09±0.0 | 0.13±0.0 | 2.08±0.0 | 13.64±3.6 | 7.86±1.8 |
| | Ours | 0.01±0.0 | 0.03±0.0 | 0.01±0.0 | 0.01±0.0 | 0.05±0.0 | 0.01±0.0 | 0.04±0.0 | 0.02±0.0 | 0.03±0.0 | 0.01±0.0 | 0.02±0.0 | 2.09±0.0 | 3.47±0.3 | 2.78±0.2 |
| Axis Pos | Ditto | 3.77 | 1.02 | 0.01 | 0.13 | 5.70 | 0.20 | 5.41 | 0.66 | - | - | 2.11 | 1.84 | - | 1.84 |
| | PARIS* | 0.25±0.5 | 1.13±2.6 | 0.00±0.0 | 0.05±0.2 | 1.59±1.7 | 4.67±3.9 | 3.35±3.1 | 3.28±3.1 | - | - | 1.79±1.5 | 0.50±0.0 | - | 0.50±0.0 |
| | CSG-reg | 0.02±0.0 | 0.00±0.0 | 0.20±0.2 | 0.18±0.0 | 0.01±0.0 | 0.02±0.0 | 0.01±0.0 | 2.13±1.5 | - | - | 0.32±0.2 | 1.46±1.1 | - | 1.46±1.1 |
| | 3Dseg-reg | - | - | 0.10±0.0 | - | - | - | - | - | - | - | - | - | - | - |
| | DTA | 0.01±0.0 | 0.01±0.0 | 0.01±0.0 | 0.01±0.0 | 0.02±0.0 | 0.02±0.0 | 0.00±0.0 | 0.05±0.0 | - | - | 0.02±0.0 | 0.59±0.0 | - | 0.59±0.0 |
| | Ours | 0.00±0.0 | 0.00±0.0 | 0.01±0.0 | 0.00±0.0 | 0.01±0.0 | 0.00±0.0 | 0.00±0.0 | 0.00±0.0 | - | - | 0.00±0.0 | 0.47±0.0 | - | 0.47±0.0 |
| Part Motion | Ditto | 99.36 | F | 5.18 | 2.09 | 19.28 | 56.61 | 80.60 | 55.72 | F | 0.09 | 39.87 | 8.43 | 0.38 | 4.41 |
| | PARIS* | 127.34±75.0 | 45.26±58.5 | 0.03±0.0 | 9.13±28.8 | 68.36±64.8 | 107.76±68.1 | 96.93±67.8 | 49.77±26.5 | 0.36±0.2 | 0.30±0.0 | 50.52±39.0 | 1.58±0.0 | 0.57±0.1 | 1.07±0.1 |
| | CSG-reg | 0.13±0.0 | 0.29±0.0 | 0.35±0.0 | 0.58±0.0 | 0.20±0.0 | 0.44±0.0 | 10.48±9.3 | 158.99±8.8 | 0.05±0.0 | 0.04±0.0 | 17.16±1.8 | 14.82±0.1 | 0.64±0.1 | 7.73±0.1 |
| | 3Dseg-reg | - | - | 1.61±0.1 | - | - | - | - | - | 0.15±0.0 | - | - | - | - | - |
| | DTA | 0.10±0.0 | 0.12±0.0 | 0.11±0.0 | 0.12±0.0 | 0.37±0.0 | 0.08±0.0 | 0.15±0.0 | 0.28±0.1 | 0.00±0.0 | 0.00±0.0 | 0.13±0.0 | 1.85±0.0 | 0.14±0.0 | 1.00±0.0 |
| | Ours | 0.03±0.0 | 0.04±0.0 | 0.02±0.0 | 0.02±0.0 | 0.04±0.0 | 0.01±0.0 | 0.03±0.0 | 0.03±0.0 | 0.00±0.0 | 0.00±0.0 | 0.02±0.0 | 1.94±0.0 | 0.04±0.0 | 0.99±0.0 |
| CD-s | Ditto | 33.79 | 3.05 | 0.25 | 2.52 | 39.07 | 41.64 | 2.64 | 10.32 | 46.90 | 9.18 | 18.94 | 47.01 | 16.09 | 31.55 |
| | PARIS* | 10.20±5.8 | 8.82±12.0 | 0.16±0.0 | 3.18±0.3 | 15.58±13.3 | 2.48±1.9 | 1.95±5.5 | 12.19±3.7 | 1.40±0.7 | 8.67±0.8 | 6.46±3.9 | 11.64±5.0 | 20.25±2.8 | 15.94±2.1 |
| | CSG-reg | 1.69 | 1.45 | 0.32 | 3.93 | 3.26 | 2.22 | 1.95 | 4.53 | 0.59 | 7.06 | 2.70 | 6.33 | 12.55 | 9.44 |
| | 3Dseg-reg | - | - | 0.76 | - | - | - | - | - | 66.31 | - | - | - | - | - |
| | DTA | 0.18±0.0 | 0.62±0.0 | 0.30±0.0 | 4.60±0.1 | 3.55±6.1 | 2.91±0.1 | 2.32±0.1 | 4.56±0.1 | 0.55±0.0 | 4.90±8.5 | 2.45±0.7 | 2.36±0.1 | 10.98±0.1 | 6.67±0.1 |
| | Ours | 0.26±0.3 | 0.52±0.0 | 0.63±0.0 | 3.88±0.0 | 0.61±0.3 | 3.83±0.1 | 2.25±0.2 | 6.43±0.1 | 0.54±0.0 | 7.31±0.2 | 2.63±0.1 | 1.64±0.2 | 2.93±0.3 | 2.29±0.3 |
| CD-m | Ditto | 141.11 | 0.99 | 0.19 | 0.94 | 20.68 | 31.21 | 15.88 | 12.89 | 195.93 | 2.20 | 42.20 | 50.60 | 20.35 | 35.48 |
| | PARIS* | 17.97±24.9 | 7.23±11.5 | 0.15±0.0 | 6.54±10.6 | 16.65±16.6 | 30.46±37.0 | 10.17±6.9 | 265.27±248.7 | 117.99±213.0 | 52.34±11.0 | 52.48±58.0 | 77.85±26.8 | 474.57±227.2 | 276.21±127.0 |
| | CSG-reg | 1.91 | 21.71 | 0.42 | 256.99 | 1.95 | 6.36 | 29.78 | 436.42 | 26.62 | 1.39 | 78.36 | 442.17 | 521.49 | 481.83 |
| | 3Dseg-reg | - | - | 1.01 | - | - | - | - | - | 6.23 | - | - | - | - | - |
| | DTA | 0.15±0.0 | 0.27±0.0 | 0.13±0.0 | 0.44±0.0 | 10.11±19.4 | 1.13±0.5 | 1.47±0.0 | 2.05±0.3 | 0.45±0.2 | 2.05±0.3 | 1.66±2.0 | 1.12±0.0 | 30.78±2.6 | 15.95±1.3 |
| | Ours | 0.54±0.1 | 0.21±0.0 | 0.13±0.0 | 0.89±0.0 | 0.64±0.0 | 0.52±0.0 | 1.22±0.0 | 0.45±0.2 | 1.12±0.0 | 1.02±0.4 | 0.67±0.2 | 0.66±0.2 | 6.28±3.6 | 3.47±1.9 |
| CD-w | Ditto | 6.80 | 2.16 | 0.31 | 2.51 | 1.70 | 2.38 | 2.09 | 7.29 | 42.04 | 3.91 | 7.12 | 6.50 | 14.08 | 10.29 |
| | PARIS* | 4.37±6.4 | 5.53±4.7 | 0.26±0.0 | 3.18±0.3 | 3.90±3.6 | 5.27±5.9 | 1.78±0.2 | 10.11±1.4 | 0.58±0.0 | 7.80±0.4 | 4.28±2.4 | 8.99±1.4 | 32.10±8.2 | 20.55±4.8 |
| | CSG-reg | 0.48 | 0.98 | 0.40 | 3.00 | 1.70 | 1.99 | 1.20 | 4.48 | 0.56 | 4.00 | 1.88 | 5.71 | 14.29 | 10.00 |
| | 3Dseg-reg | - | - | 0.81 | - | - | - | - | - | 0.78 | - | - | - | - | - |
| | DTA | 0.27±0.0 | 0.70±0.0 | 0.32±0.0 | 4.24±0.1 | 0.41±0.0 | 1.92±0.0 | 1.17±0.0 | 4.48±0.2 | 0.36±0.0 | 3.99±0.4 | 1.79±0.1 | 2.08±0.1 | 8.98±0.1 | 5.53±0.1 |
| | Ours | 0.43±0.2 | 0.58±0.0 | 0.50±0.0 | 3.58±0.0 | 0.67±0.3 | 2.63±0.0 | 1.28±0.0 | 5.99±0.1 | 0.61±0.0 | 5.21±0.1 | 2.15±0.1 | 1.29±0.1 | 3.23±0.1 | 2.26±0.1 |
| Time (min) | DTA | 29 | 30 | 31 | 29 | 28 | 29 | 31 | 28 | 27 | 28 | 29 | 29 | 29 | 29 |
| | Ours | 9 | 8 | 7 | 7 | 7 | 7 | 7 | 8 | 7 | 8 | 8 | 9 | 9 | 9 |

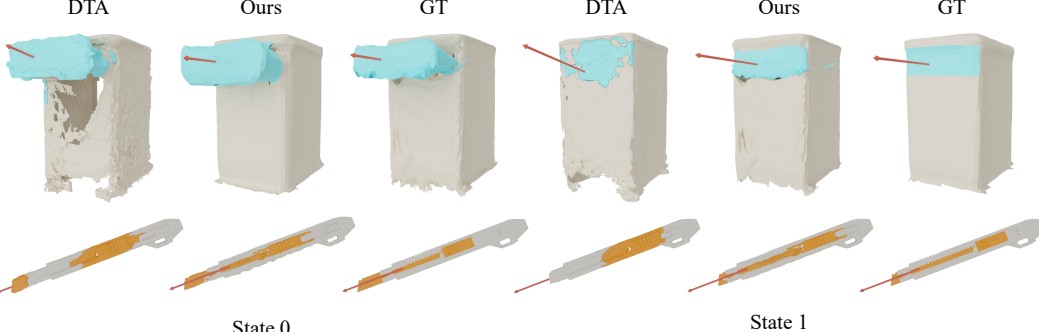

DTA    Ours    GT    DTA    Ours    GT

State 0    State 1

Figure 2: **Qualitative visualizations of PARIS objects.** We present reconstruction comparisons between DTA and our model on Real Storage (Top) and Synthetic Blade (Bottom). DTA struggles with mesh reconstruction at the low-visibility state, as it processes each state separately. In contrast, our method leverages the connection between states to improve the reconstruction for both low- and high-visibility states.

as the number of parts increases, whereas our model maintains high performance regardless of part count. We also provide a qualitative comparison in Fig. 3 for better visualization. Moreover, our method maintains the same time efficiency while the training time of existing methods scales with the number of parts. These results underscore the robustness and effectiveness of our method in modeling complex, multi-part articulated objects.

## 5.3 ABLATIVE STUDIES

**Experimental Setup** To verify the effectiveness of our model design, we meticulously design four ablations of ArtGS to identify the impact of key components in our method: (i) Randomly initializing canonical Gaussians (*w/o* Cano. Init.), (ii) predicting part assignments with MLP (*w/* MLP Seg) or Slot-Attention (*w/* SA Seg), (iii) randomly initializing part centers $C_k$ (*w/o* Center Init.), (iv)

Table 2: **Quantitative evaluation on DTA-Multi.** We report averaged metrics over 10 trials with different random seeds. Lower (↓) is better on all metrics. Joint 1 of "Storage-m" is prismatic with no Axis Pos.

| Object | Method | Axis Ang 0 | Axis Ang 1 | Axis Pos 0 | Axis Pos 1 | Part Motion 0 | Part Motion 1 | CD-s | CD-m 0 | CD-m 1 | CD-w | Time (min) |
|---|---|---|---|---|---|---|---|---|---|---|---|---|
| | PARIS | 34.52 | 15.91 | 3.60 | 1.63 | 86.21 | 105.86 | 8.52 | 526.19 | 160.86 | 15.00 | - |
| Fridge-m | DTA | 0.25 | 0.06 | 0.01 | 0.01 | 0.23 | 0.08 | 0.63 | 0.44 | 0.53 | 0.88 | 32 |
| | Ours | **0.02** | **0.00** | **0.00** | **0.00** | **0.02** | **0.03** | **0.62** | **0.07** | **0.18** | **0.75** | **8** |
| | PARIS | 43.26 | 26.18 | 10.42 | - | 79.84 | 0.64 | 8.56 | 128.62 | 266.71 | 8.66 | - |
| Storage-m | DTA | 0.17 | 0.40 | 0.04 | - | 0.13 | 0.00 | 0.86 | 0.20 | **0.25** | 0.97 | 32 |
| | Ours | **0.01** | **0.02** | **0.01** | - | **0.01** | **0.00** | **0.78** | **0.19** | 0.27 | **0.93** | **8** |

Table 3: **Quantitative evaluation on ArtGS-Multi**. Metrics are averaged over 3 trials. Due to the large number of parts, we report the average metric for all movable parts. Lower (↓) is better on all metrics. "Table-31249" has 3 prismatic joints with no Axis Pos.

| Object | Method | Axis Ang | Axis Pos | Part Motion | CD-s | CD-m | CD-w | Time (min) |
|---|---|---|---|---|---|---|---|---|
| Table | DTA | 24.35 | - | 0.12 | **0.59** | 104.38 | **0.55** | 34 |
| 25493 (4 parts) | Ours | **1.16** | - | **0.00** | 0.74 | **3.53** | 0.74 | **8** |
| Table | DTA | 20.62 | 4.2 | 30.8 | 1.39 | 230.38 | **1.00** | 37 |
| 31249 (5 parts) | Ours | **0.04** | **0.00** | **0.01** | **1.22** | **3.09** | 1.16 | **8** |
| Storage | DTA | 51.18 | 2.44 | 43.77 | 5.74 | 246.63 | 0.88 | 35 |
| 45503 (4 parts) | Ours | **0.02** | **0.00** | **0.03** | **0.75** | **0.13** | 0.88 | **8** |
| Storage | DTA | 19.07 | 0.31 | 10.67 | 0.82 | 476.91 | 0.71 | 45 |
| 47468 (7 parts) | Ours | **0.14** | **0.02** | **0.62** | **0.67** | **3.70** | **0.70** | **8** |
| Oven | DTA | 17.83 | 6.51 | 31.80 | 1.17 | 359.16 | **1.01** | 35 |
| 101908 (4 parts) | Ours | **0.04** | **0.01** | **0.23** | **1.08** | **0.25** | 1.03 | **8** |

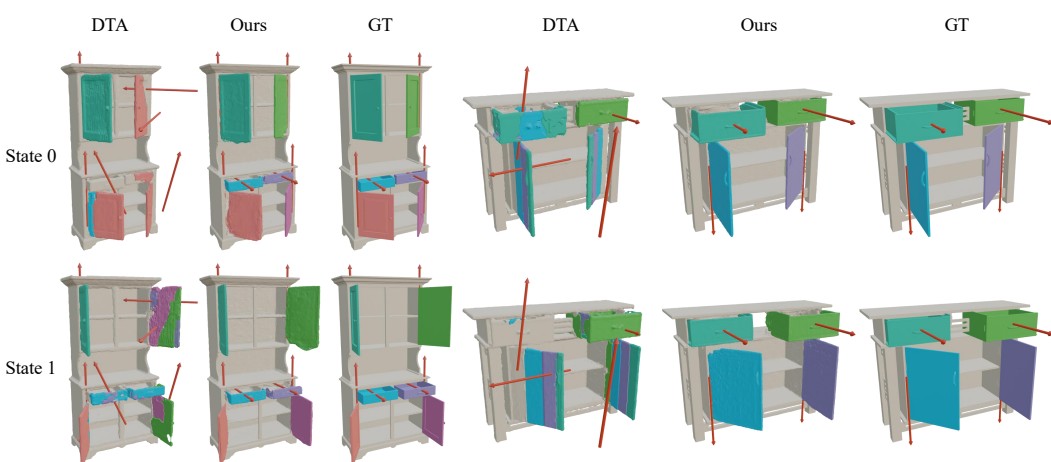

Figure 3: **Qualitative results on multi-part objects.** We present reconstruction comparisons between DTA and our model on Storage-47648 (Left) and Table-31249 (Bottom). On ArtGS-Multi, DTA struggles with movable part identification and axis prediction as the number of parts increases, whereas our model maintains high performance regardless of part count, achieving high-quality reconstruction of part mesh and joint articulation.

clustering all Gaussians instead of clustering movable Gaussians for part center initialization (*w/o Motion Prior*), and (v) learning articulation parameters without the joint prediction warmup stage (*w/o Joint Pred.*). We select two representative objects: "Storage-47648" with 4 revolute and 2 prismatic joints and "Oven-101908" with 3 revolute joints for ablative analysis. Similar to Sec. 5.2, we report the average of all parts over 10 trials for all metrics.

**Results and Discussions** As shown in Tab. 4 and Fig. 4, we make the following observations:

- *Canonical Gaussians Initialization.* Omitting this initialization strategy significantly degrades the model performance across all metrics, particularly for movable parts. As illustrated in Fig. 4

Table 4: **Ablative experiments**. Lower (↓) is better on all metrics.

| Method | Storage 47648 (7 parts) | | | | | | Oven 101908 (4 parts) | | | | | |
|---|---|---|---|---|---|---|---|---|---|---|---|---|
| | Axis Ang | Axis Pos | Part Motion | CD-s | CD-m | CD-w | Axis Ang | Axis Pos | Part Motion | CD-s | CD-m | CD-w |
| Full | **0.14** | **0.02** | **0.62** | **0.67** | **3.70** | **0.70** | **0.04** | **0.01** | **0.23** | **1.08** | **0.25** | **1.03** |
| *w/o* Cano. init. | 24.15 | 0.73 | 20.61 | 0.83 | 495.07 | 1.25 | 57.87 | 2.95 | 54.45 | 1.73 | 1030.19 | 2.36 |
| *w/o* Center Init. | 52.78 | 0.83 | 33.04 | 1.09 | 344.19 | 1.69 | 28.94 | 2.36 | 22.46 | 1.41 | 8.86 | 2.13 |
| *w/o* Motion Prior | 26.74 | 0.22 | 21.16 | 258.23 | 599.46 | 1.15 | 40.08 | 0.98 | 41.06 | 1.75 | 503.44 | 2.35 |
| *w/o* Joint Pred. | 0.16 | 0.02 | 0.72 | 0.67 | 3.90 | 0.71 | 0.04 | 0.01 | 0.23 | 1.08 | 0.25 | 1.03 |
| *w/* MLP Seg | 21.84 | 3.46 | 31.43 | 1.82 | 664.25 | 1.28 | 12.08 | 3.33 | 27.28 | 7.78 | 126.95 | 2.19 |
| *w/* SA Seg | 25.43 | 0.7 | 23.22 | 1.52 | 459.89 | 1.16 | 58.04 | 4.53 | 51.28 | 1.26 | 496.64 | 2.35 |

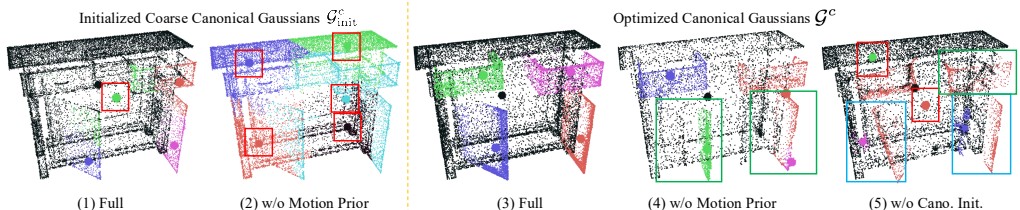

Figure 4: **Abaltion Studies**. We visualize the initialized and optimized canonical Gaussians with their part assignment and centers for the full model, w/o Motion Prior and w/o Cano. Init. We highlight center error, part assignment error, and canonical Gaussian error with red, green, and blue bounding boxes separately.

(5), the absence of our initialization strategy leads to malformed canonical Gaussians, making the model converge to suboptimal local minima during optimization.

- *Center-based Part Modeling and Assignment.* Replacing our center-based part assignment module with MLP or Slot-Attention ("w/ MLP Seg" and "w/ SA Seg") leads to substantial performance drops, especially in joint parameter estimation and movable part reconstruction. This demonstrates the superiority of our center-based approach in accurately segmenting articulated parts.
- *Center Initialization.* Random center initialization performs well for static parts but poorly for movable parts. Clustering all Gaussians fails to reconstruct both static and movable parts due to incorrect center initialization. As illustrated in Fig. 4 (1), clustering on movable Gaussians still produces an incorrect center but provides a good starting point for optimization. Our ArtGS will refine the centers in the optimization process as shown in Fig. 4 (3). In contrast, clustering on all Gaussians results in entirely wrong center initialization (Fig. 4 (2)), which is difficult to correct (Fig. 4 (4)), leading to even worse performance than random initialization. This highlights the importance of our center initialization strategy in achieving accurate part articulation modeling.
- *Joint Prediction Warmup.* This technique primarily affects prismatic joints, as we do not constrain the transformation of revolute joints. As shown in Tab. 4, predicting the joint type and then refining joint parameters with type constraints slightly improves the articulation reconstruction.

In summary, these ablation studies confirm that each component contributes significantly to its overall performance, playing crucial roles in achieving accurate joint parameter estimation and high-quality part mesh reconstruction. We provide further discussions in Appendix B and Appendix C.

## 6 CONCLUSION

In conclusion, we propose ArtGS, a novel approach for reconstructing articulated objects from two states of multi-view images. By leveraging 3D Gaussians and introducing novel techniques for state alignment and part dynamics modeling, our approach overcomes key limitations of existing methods. The performance improvements in joint parameter estimation and part mesh reconstruction, particularly for complex multi-part objects, demonstrate the effectiveness of our innovations. Our comprehensive experiments across synthetic and real-world datasets validate the robustness and efficiency of ArtGS, while also revealing promising directions for future research. As the demand for accurate digital replicas of articulated objects continues to grow in fields such as robotics and augmented reality, ArtGS provides a solid foundation for bridging the gap between physical and virtual environments. Moving forward, we anticipate that the principles introduced in this work will inspire further advancements in the field, ultimately enabling more sophisticated and realistic simulations for a wide range of applications.

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

## A  IMPLEMENTATION AND TRAINING DETAILS

**Canonical Gaussian Initialzation**  We train single-state Gaussians $\mathcal{G}^0$ and $\mathcal{G}^1$ for 10K steps with loss $\mathcal{L} = (1 - \lambda_{\text{SSIM}})\mathcal{L}_I + \lambda_{\text{SSIM}}\mathcal{L}_{\text{D-SSIM}} + \lambda_o \mathcal{L}_o$, where $\lambda_{\text{SSIM}} = 0.2$, $\lambda_o = 0.01$ is used in experiments and $\mathcal{L}_o$ is an opacity entropy loss calculated as:

$$\hat{\sigma}_i = \mathbb{1}\{\sigma_i > 0.5\}, \quad \mathcal{L}_o = -\frac{1}{N}\sum_{i=1}^{N}[\hat{\sigma}_i \sigma_i + (1 - \hat{\sigma}_i)\log(1 - \sigma_i)],$$

which encourages Gaussian opacities $\sigma_i$ to approach either 0 or 1, controlling Gaussian count and accelerating training. We then obtain coarse canonical Gaussians by matching $\mathcal{G}^0$ and $\mathcal{G}^1$ as described in Sec. 4.1. This stage takes about 2 minutes per object.

**Part Discovery for Articulation Modeling**  As described in Sec. 4.2, given canonical Gaussians $\mathcal{G}^c = \{G_i\}_{i=1}^{N}$ and $K$ learnable part centers $C_k = (\boldsymbol{p}_k, \boldsymbol{R}_k, \boldsymbol{\lambda}_k)$, we calculate part-level masks $\boldsymbol{M}$ using Eq. (6). We use a learnable hash grid $H$ to encode Gaussian positions and predict the residual term in Eq. (6) as:

$$\mathbf{X}_i^k = \frac{[\boldsymbol{R}_k(\boldsymbol{\mu}_i^c - \boldsymbol{p}_k)]}{\boldsymbol{\lambda}_k}, \quad \mathbf{D}_{ik} = (\mathbf{X}_i^k)^T \cdot \mathbf{X}_i^k$$

$$\boldsymbol{W}_{\Delta ik} = \text{MLP}(\boldsymbol{\mu}_i^c, H(\boldsymbol{\mu}_i^c), \{X_i^k\}_{k=1}^K, \{D_{ik}\}_{k=1}^K), \quad \boldsymbol{M} = \text{GumbelSoftmax}\left(\frac{-\boldsymbol{D} + \boldsymbol{W}_\Delta}{\tau}\right)$$

Since the part assignment and articulation parameters are far from optimal at the beginning of training, using hard assignment for Gumbel-Softmax hinders the joint optimization of the part assignment and articulation parameters. To address this problem, we anneal the temperature $\tau$ from 1 to 0.1 over 10K steps, using soft assignment that is similar to Softmax when $\tau > 0.1$ and hard assignment otherwise for training stability. This approach allows for more flexible assignments during the early stages of training, facilitating better joint optimization, and gradually transitioning to decisive part assignment as the model converges.

**Optimization**  To enhance the learning of articulation parameters, we adopt a warm-up strategy to predict the joint type of each part. This process requires 3K-5K steps that take 30 to 50 seconds. Then we train ArtGS with joint type constraint for 20K steps, taking 5-7 minutes per object. For hyper-parameters, we set the threshold $\epsilon_{\text{static}}$ to identify static/movable Gaussians as $\epsilon_{\text{static}} = 0.02 \cdot \max_i \text{CD}_i^{t \to \bar{t}}$ for two-part objects and $\epsilon_{\text{static}} = 0.05 \cdot \max_i \text{CD}_i^{t \to \bar{t}}$ for multi-part objects. We use $\epsilon_{\text{revol}} = 10°$ for predicting joint types following PARIS (Liu et al., 2023a). $\lambda_{cd}$ and $\lambda_{reg}$ are set as 100 and 0.1 separately. In addition, the CD loss in Eq. (9) aims to decrease the distance between a deformed Gaussian $G_i^t$ and its nearest Gaussians $\hat{G}_i^t$ in $\mathcal{G}_{\text{single}}^t$. Since the deformed Gaussians and canonical Gaussians for a prismatic joint have a large overlap, the nearest Gaussian may be in the opposite direction of the ideal one, making it ineffective for prismatic joints. Thus the CD loss is only used for regularizing the objects that only have revolute joints. Moreover, the densification strategy of Gaussians is cloning or splitting one Gaussian when the gradient of its center $\boldsymbol{\mu}$ is greater than a threshold $\epsilon_{\text{densify}}$. This is effective for static scenes but meets challenges for dynamic scenes. In the early stage of training, the large gradient is often due to deformation error. To prevent excessive increase of Gaussian quantity due to deformation error, we raised this threshold $\epsilon_{\text{densify}}$ from 0.0002 used in previous works (Kerbl et al., 2023; Huang et al., 2024c) to 0.001.

## B  ADDITIONAL DISCUSSIONS

We present a comprehensive analysis of ArtGS and DTA through additional quantitative and qualitative results.

**Visibility Problem**  Our results uncover an intriguing inconsistency in DTA's performance across different states of the same object. As illustrated in Tab. A.1, DTA demonstrates good reconstruction quality in the high-visibility state but shows markedly poor performance in the low-visibility state. This limitation is particularly pronounced for objects with prismatic joints, such as real storage and blade. In these cases, DTA struggles to accurately capture the geometry and articulation of partially

Table A.1: **Quantitative evaluation of each state on PARIS data.** We report the average of metrics over 10 trials of each state. "metric-0/1" represents the metric evaluated at state 0/1 and "metric-m" is the average of two states. We highlight best results on average of two states. Axis Pos. is omitted for prismatic joints (Blade, Storage, and Real Storage).

| Metric | Method | Synthetic Objects | | | | | | | | | | | Real Objects | | |
|---|---|---|---|---|---|---|---|---|---|---|---|---|---|---|---|
| | | FoldChair | Fridge | Laptop | Oven | Scissor | Stapler | USB | Washer | Blade | Storage | All | Fridge | Storage | All |
| Axis Ang | DTA-0 | 0.03 | 0.09 | 0.07 | 0.22 | 0.10 | 0.06 | 0.11 | 0.36 | 0.20 | 0.07 | 0.13 | 2.08 | 13.64 | 7.86 |
| | Ours-0 | 0.01 | 0.03 | 0.01 | 0.01 | 0.05 | 0.01 | 0.04 | 0.02 | 0.03 | 0.01 | 0.02 | 2.09 | 3.47 | 2.78 |
| | DTA-1 | 0.04 | 0.10 | 0.07 | 0.23 | 0.10 | 0.07 | 0.11 | 0.36 | 0.26 | 0.09 | 0.14 | 2.07 | 8.08 | 5.08 |
| | Ours-1 | 0.01 | 0.03 | 0.01 | 0.01 | 0.05 | 0.01 | 0.04 | 0.02 | 0.03 | 0.01 | 0.02 | 2.09 | 3.47 | 2.78 |
| | DTA-m | 0.04 | 0.10 | 0.07 | 0.22 | 0.10 | 0.06 | 0.11 | 0.36 | 0.23 | 0.08 | 0.14 | **2.08** | 10.86 | 6.47 |
| | Ours-m | **0.01** | **0.03** | **0.01** | **0.01** | **0.05** | **0.01** | **0.04** | **0.02** | **0.03** | **0.01** | **0.02** | 2.09 | **3.47** | **2.78** |
| Axis Pos | DTA-0 | 0.01 | 0.01 | 0.01 | 0.01 | 0.03 | 0.02 | 0.00 | 0.04 | - | - | 0.02 | 0.59 | - | 0.59 |
| | Ours-0 | 0.00 | 0.00 | 0.01 | 0.00 | 0.00 | 0.01 | 0.00 | 0.00 | - | - | 0.00 | 0.47 | - | 0.47 |
| | DTA-1 | 0.01 | 0.01 | 0.01 | 0.01 | 0.02 | 0.02 | 0.00 | 0.05 | - | - | 0.02 | 0.59 | - | 0.59 |
| | Ours-1 | 0.00 | 0.00 | 0.01 | 0.00 | 0.00 | 0.01 | 0.00 | 0.00 | - | - | 0.00 | 0.47 | - | 0.47 |
| | DTA-m | 0.01 | 0.01 | 0.01 | 0.01 | 0.03 | 0.02 | 0.00 | 0.04 | - | - | 0.02 | 0.59 | - | 0.59 |
| | Ours-m | **0.00** | **0.00** | **0.01** | **0.00** | **0.00** | **0.01** | **0.00** | **0.00** | **-** | **-** | **0.00** | **0.47** | **-** | **0.47** |
| Part Motion | DTA-0 | 0.10 | 0.12 | 0.11 | 0.12 | 0.38 | 0.08 | 0.15 | 0.28 | 0.00 | 0.00 | 0.13 | 1.85 | 0.14 | 1.00 |
| | Ours-0 | 0.03 | 0.04 | 0.02 | 0.02 | 0.04 | 0.01 | 0.03 | 0.03 | 0.00 | 0.00 | 0.02 | 1.94 | 0.04 | 0.99 |
| | DTA-1 | 0.09 | 0.13 | 0.11 | 0.13 | 0.37 | 0.08 | 0.14 | 0.28 | 0.00 | 0.00 | 0.13 | 1.85 | 0.09 | 0.97 |
| | Ours-1 | 0.03 | 0.04 | 0.02 | 0.02 | 0.04 | 0.01 | 0.03 | 0.03 | 0.00 | 0.00 | 0.02 | 1.94 | 0.04 | 0.99 |
| | DTA-m | 0.09 | 0.12 | 0.11 | 0.12 | 0.38 | 0.08 | 0.15 | 0.28 | 0.00 | 0.00 | 0.13 | **1.85** | 0.12 | 0.99 |
| | Ours-m | **0.03** | **0.04** | **0.02** | **0.02** | **0.04** | **0.01** | **0.03** | **0.03** | **0.00** | **0.00** | **0.02** | 1.94 | **0.04** | **0.99** |
| CD-s | DTA-0 | 0.18 | 0.62 | 0.32 | 4.60 | 3.30 | 2.68 | 2.32 | 4.77 | 0.55 | 4.71 | 2.41 | 2.36 | 10.98 | 6.67 |
| | Ours-0 | 0.26 | 0.52 | 0.59 | 3.88 | 0.62 | 3.85 | 2.25 | 6.41 | 0.54 | 7.47 | 2.64 | 1.64 | 2.93 | 2.29 |
| | DTA-1 | 0.19 | 0.63 | 0.30 | 4.58 | 3.55 | 2.91 | 2.90 | 4.56 | 0.45 | 4.90 | 2.50 | 2.59 | 9.60 | 6.10 |
| | Ours-1 | 0.26 | 0.48 | 0.63 | 4.00 | 0.61 | 3.83 | 2.56 | 6.43 | 0.54 | 7.31 | 2.67 | 2.01 | 4.02 | 3.02 |
| | DTA-m | **0.19** | 0.62 | **0.31** | 4.59 | 3.43 | **2.79** | 2.61 | **4.66** | **0.50** | **4.80** | **2.46** | 2.48 | 10.29 | 6.39 |
| | Ours-m | 0.26 | **0.50** | 0.61 | **3.94** | **0.61** | 3.84 | **2.41** | 6.42 | 0.54 | 7.39 | 2.65 | **1.82** | **3.48** | **2.65** |
| CD-m | DTA-0 | 0.15 | 0.27 | 0.16 | 0.44 | 17.38 | 2.34 | 1.47 | 0.37 | 2.05 | 0.36 | 2.50 | 1.12 | 30.78 | 15.95 |
| | Ours-0 | 0.54 | 0.21 | 0.14 | 0.89 | 0.65 | 0.88 | 1.22 | 1.54 | 1.12 | 1.03 | 0.82 | 0.66 | 6.28 | 3.47 |
| | DTA-1 | 0.13 | 0.30 | 0.13 | 0.45 | 10.11 | 1.13 | 1.51 | 0.45 | 61.38 | 0.36 | 7.60 | 1.85 | 365.74 | 183.80 |
| | Ours-1 | 0.12 | 0.21 | 0.13 | 0.76 | 0.64 | 0.52 | 1.43 | 0.45 | 1.01 | 1.02 | 0.63 | 1.31 | 87.81 | 44.56 |
| | DTA-m | **0.14** | 0.28 | 0.15 | **0.44** | 13.75 | 1.73 | 1.49 | **0.41** | 31.72 | **0.36** | 5.05 | 1.48 | 198.26 | 99.88 |
| | Ours-m | 0.33 | **0.21** | **0.14** | 0.82 | **0.65** | **0.70** | **1.33** | 1.00 | **1.06** | 1.02 | **0.73** | **0.99** | **47.05** | **24.02** |
| CD-w | DTA-0 | 0.27 | 0.70 | 0.35 | 4.24 | 0.42 | 2.13 | 1.17 | 4.59 | 0.36 | 4.09 | 1.83 | 2.08 | 8.98 | 5.53 |
| | Ours-0 | 0.43 | 0.58 | 0.47 | 3.58 | 0.69 | 3.13 | 1.28 | 6.12 | 0.61 | 5.13 | 2.20 | 1.29 | 3.23 | 2.26 |
| | DTA-1 | 0.26 | 0.70 | 0.32 | 4.27 | 0.41 | 1.92 | 1.52 | 4.48 | 0.38 | 3.99 | 1.83 | 2.19 | 9.03 | 5.61 |
| | Ours-1 | 0.30 | 0.59 | 0.50 | 3.71 | 0.67 | 2.63 | 1.87 | 5.99 | 0.65 | 5.21 | 2.21 | 1.45 | 2.45 | 1.95 |
| | DTA-m | **0.26** | 0.70 | **0.34** | 4.25 | **0.41** | 2.02 | **1.34** | **4.53** | **0.37** | **4.04** | **1.83** | 2.13 | 9.01 | 5.57 |
| | Ours-m | 0.36 | **0.59** | 0.48 | **3.64** | 0.68 | 2.88 | 1.58 | 6.05 | 0.63 | 5.17 | 2.21 | **1.37** | **2.84** | **2.11** |

occluded parts. The observed inconsistency and state-dependent performance fluctuations underscore the necessity for a more robust approach that effectively connects and leverages information from multiple states. This is precisely where ArtGS's strengths become evident. By establishing connections between different articulation states, ArtGS achieves more consistent and high-quality reconstructions across varying object configurations. Jointly optimizing over multiple states allows ArtGS to: 1) Leverage complementary information from different articulation states, 2) Maintain consistency in part assignment and geometry across states, 3) Better handle occlusions and low-visibility scenarios by inferring occluded geometries from other states. These capabilities enable ArtGS to produce more accurate and reliable reconstructions, particularly in challenging scenarios. The superior performance of ArtGS demonstrates its potential for robust articulated object reconstruction in real-world applications.

**Significance of Part Assignment** Through analysis of both qualitative (Fig. A.3) and quantitative (Tab. 3) results, we have identified that the model's ultimate performance is primarily determined by the accuracy of part assignment. When the model fails to correctly divide an object into parts, it becomes impossible to obtain reasonable joint parameter estimation. Conversely, even when joint parameter estimation is inaccurate, the model may still correctly separate the object's parts. This insight reveals that accurate part assignment is a crucial prerequisite for high-quality articulated object reconstruction. Our findings emphasize that to enhance the reconstruction of articulated objects, the ability to reasonably separate parts is of paramount importance. ArtGS addresses this challenge through the center-based segmentation and improved initialization by clustering. These techniques work in synergy to significantly improve the part segmentation capabilities of ArtGS. By enhancing the model's ability to correctly identify and separate object parts, we lay a solid foundation for

Table A.2: **Quantitative evaluation of Axis Pos metric on PARIS.** Metrics are reported as mean $\pm$ std over 10 trials on average of 2 states. We report the value timed by 1000 and highlight the best results.

| Metric | Method | FoldChair | Fridge | Laptop | Oven | Scissor | Stapler | USB | Washer | All |
|--------|--------|-----------|--------|--------|------|---------|---------|-----|--------|-----|
| Axis Pos | DTA | $0.53_{\pm0.3}$ | $0.62_{\pm0.3}$ | $1.10_{\pm0.7}$ | $1.49_{\pm1.0}$ | $2.48_{\pm2.8}$ | $2.21_{\pm1.8}$ | $0.35_{\pm0.2}$ | $4.53_{\pm2.8}$ | $1.66_{\pm1.2}$ |
|  | Ours | $\mathbf{0.48_{\pm0.2}}$ | $\mathbf{0.44_{\pm0.2}}$ | $\mathbf{0.39_{\pm0.3}}$ | $\mathbf{0.55_{\pm0.4}}$ | $\mathbf{0.16_{\pm0.1}}$ | $\mathbf{0.93_{\pm0.4}}$ | $\mathbf{0.08_{\pm0.1}}$ | $\mathbf{0.33_{\pm0.3}}$ | $\mathbf{0.42_{\pm0.3}}$ |

subsequent stages of the reconstruction process, including joint parameter estimation and final mesh reconstruction.

## C  LIMITATIONS

**Stability of Randomness.**   ArtGS exhibits enhanced robustness and stability across different random seeds, primarily due to our innovative initialization strategy for canonical Gaussians and our part assignment module. We observe that stability issues often stem from the initialization of three key components: canonical Gaussians $\mathcal{G}^c$, part centers $C$ in the part assignment module, and joint articulation parameters $\Psi$. As demonstrated in Sec. 5.3, faulty initialization of $\mathcal{G}^c$ and $C$ can lead to significant performance degradation, particularly for complex objects with multiple movable parts. While our current initialization strategy has greatly improved stability, severe initialization errors in center $C$ may still result in part mis-segmentation. We can integrate prior models such as SAM (Kirillov et al., 2023) to enhance the ability to correct center initialization errors. Although ArtGS works with randomly initialized $\Psi$, we have observed that improved initialization of $\Psi$ brings enhanced performance. Future work could explore the integration of heuristic algorithms or feed-forward articulation estimation models to provide better initial estimation for $\Psi$.

**Limited States**   Our current approach is limited to modeling articulated objects using only two states, which may not fully capture the complexity of real-world multi-part objects. Moreover, as the number of parts increases, distinguishing parts with similar joint axes and motion patterns (such as parallel drawers) becomes increasingly challenging, complicating the segmentation process. To address this, two main avenues could be explored for future research: 1) Multi-state Extension: Develop a methodology to extend ArtGS to handle multiple states that interact with different parts, potentially by identifying movable parts with a sequential state update mechanism. This would involve iteratively updating the model as new state information becomes available, allowing for a more comprehensive representation of the object's articulation space. 2) Continuous Temporal Reconstruction: Adapt ArtGS to reconstruct articulated objects from monocular video sequences. This approach would leverage temporal information to infer a continuous range of articulation states, providing a more nuanced understanding of the object's movement capabilities.

**Mesh Reconstruction Fidelity**   Our current implementation utilizes the original Gaussian Splatting technique, which, while effective, has limitations in terms of mesh reconstruction quality compared with NeRF-based methods(Wang et al., 2021; Yariv et al., 2021; Wen et al., 2023). Integrating recent advancements in reconstruction with Gaussian Splatting (Huang et al., 2024a; Chen et al., 2024a) may help to improve the reconstruction fidelity of ArtGS.

## D  ADDITIONAL EXPERIMENTS

### D.1  ADDITIONAL QUANTITATIVE COMPARISONS

We provide additional comparisons with previous methods in this section.

**Scaled Axis Pos Metric.**   Following DTA and PARIS, we multiply the 'Axis Pos' metric by 10 in Tab. 1 and Tab. A.1. While this metric shows minimal variation among current methods for synthetic objects, we also report the Axis Pos metric multiplied by 1000. As shown in Tab. A.2, ArtGS demonstrates superior performance compared to DTA.

**Perception-based Metrics.**   To evaluate rendering quality, we assess perception-based metrics including LPIPS Zhang et al. (2018), SSIM Wang et al. (2004), and PSNR, with results shown

Table A.3: **Quantitative evaluation for perception-based metrics on PARIS data.** We report the results on average of two states. We highlight best results.

| Metric | Method | Synthetic Objects | | | | | | | | | | | Real Objects | | |
|---|---|---|---|---|---|---|---|---|---|---|---|---|---|---|---|
| | | FoldChair | Fridge | Laptop | Oven | Scissor | Stapler | USB | Washer | Blade | Storage | All | Fridge | Storage | All |
| PSNR | PARIS | 31.50 | **37.67** | **37.26** | 35.30 | **38.37** | 38.49 | 39.07 | **40.08** | 38.29 | 36.18 | 37.22 | 25.29 | **27.13** | 26.21 |
| | Ours | **34.46** | 37.11 | 34.09 | **37.06** | 38.29 | **39.13** | **39.64** | 38.50 | **41.16** | **37.24** | **37.67** | **27.05** | 25.38 | **26.22** |
| SSIM | PARIS | 0.985 | **0.994** | **0.991** | 0.980 | 0.996 | 0.995 | 0.992 | 0.991 | 0.996 | **0.993** | 0.991 | 0.898 | **0.953** | 0.926 |
| | Ours | **0.997** | 0.993 | 0.988 | **0.995** | **0.998** | **0.999** | **0.998** | **0.995** | **0.999** | 0.992 | **0.995** | **0.939** | 0.930 | **0.935** |
| LPIPS$_{vgg}$ | PARIS | 0.045 | **0.032** | **0.020** | **0.045** | 0.015 | 0.019 | 0.029 | **0.029** | 0.017 | **0.095** | **0.035** | 0.188 | **0.139** | 0.164 |
| | Ours | **0.036** | 0.041 | 0.045 | 0.054 | **0.014** | **0.011** | **0.016** | 0.052 | **0.004** | 0.097 | 0.037 | **0.114** | 0.188 | **0.151** |

Table A.4: **Quantitative comparison for whole mesh reconstruction on PARIS data.** We report the average of CD-w over 10 trials. We bold **best** results on average of two states.

| Metric | Method | Synthetic Objects | | | | | | | | | | | Real Objects | | |
|---|---|---|---|---|---|---|---|---|---|---|---|---|---|---|---|
| | | FoldChair | Fridge | Laptop | Oven | Scissor | Stapler | USB | Washer | Blade | Storage | All | Fridge | Storage | All |
| CD-w | DTA | **0.26** | 0.70 | **0.34** | 4.25 | **0.41** | **2.02** | **1.34** | **4.53** | **0.37** | 4.04 | **1.83** | 2.13 | 9.01 | 5.57 |
| | TSDF with gt depth | 0.30 | **0.56** | 0.47 | **3.60** | 0.49 | 2.78 | 1.60 | 5.73 | 0.54 | 5.13 | 2.12 | 3.15 | 131.86 | 67.51 |
| | Ours | 0.36 | 0.59 | 0.48 | 3.64 | 0.68 | 2.88 | 1.58 | 6.05 | 0.63 | 5.17 | 2.21 | **1.37** | **2.84** | **2.11** |

in Tab. A.3. While our primary focus aligns with previous methods on mesh reconstruction and articulation estimation, ArtGS achieves comparable or superior performance relative to PARIS.

**Limited Improvement for CD-w on Simple Synthetic Objects.** Our method's performance on simple synthetic objects, particularly in terms of CD-w metric, is constrained by our use of TSDF for mesh extraction from Gaussian Splatting-rendered depths. To analyze this limitation, we compare against meshes reconstructed using ground-truth depth with TSDF. As shown in Tab. A.4, even with ground-truth depth input, TSDF-based reconstruction cannot surpass algorithms using marching cubes with NeRF, primarily due to the fundamental differences between TSDF and marching cubes algorithms on simple geometries. However, for complex or real-world objects where articulation reconstruction becomes more critical, the advantages of our model become evident. Additionally, TSDF with ground truth depth on real-world objects may produce poor-quality meshes (e.g., real_storage) due to depth sensor noise, while our ArtGS achieves high-quality reconstruction. Importantly, our primary objective is to create digital twins of real-world articulated objects, where ArtGS demonstrates significant performance improvements, particularly for complex and real-world scenarios.

## D.2 Failure Cases

**Incorrect Initialization of Part Centers.** For real-world objects with multiple parts, clustering-derived part centers may be inaccurate (Fig. A.1 (a)) due to sensor noise, occlusion, and varying illumination conditions. These incorrectly initialized centers often persist through optimization, degrading performance for parts with misaligned centers (Fig. A.1 (c)). Manual correction of erroneous part centers prior to training (Fig. A.1 (b)) yields improved results (Fig. A.1 (d)). As discussed in Appendix C, incorporating prior models like SAM for automatic, accurate part center initialization remains a promising direction for future work.

**Similar Motions.** Our method exhibits limitations when handling parts with identical motion across states, as demonstrated in case 2 of Fig. A.1 where two drawers are pulled with the same distance. In such scenarios, the model tends to learn a single joint to fit both parts, failing to distinguish between the independently movable parts. As discussed in Appendix C, expanding ArtGS to incorporate additional states would provide richer motion information, potentially enabling better part separation.

## D.3 Evolution of Canonical Gaussians

We visualize the evolution of canonical Gaussians in Fig. A.2, showing both their part assignments and centers. Our initialization strategy begins with dense static Gaussians and sparse dynamic Gaussians. As training progresses, the Gaussians undergo densification while simultaneously refining their part centers and assignments. These visualization results demonstrate the effectiveness of ArtGS.

## D.4 Additional Qualitative Comparisons

We provide additional qualitative comparisons on different datasets in the following pages.

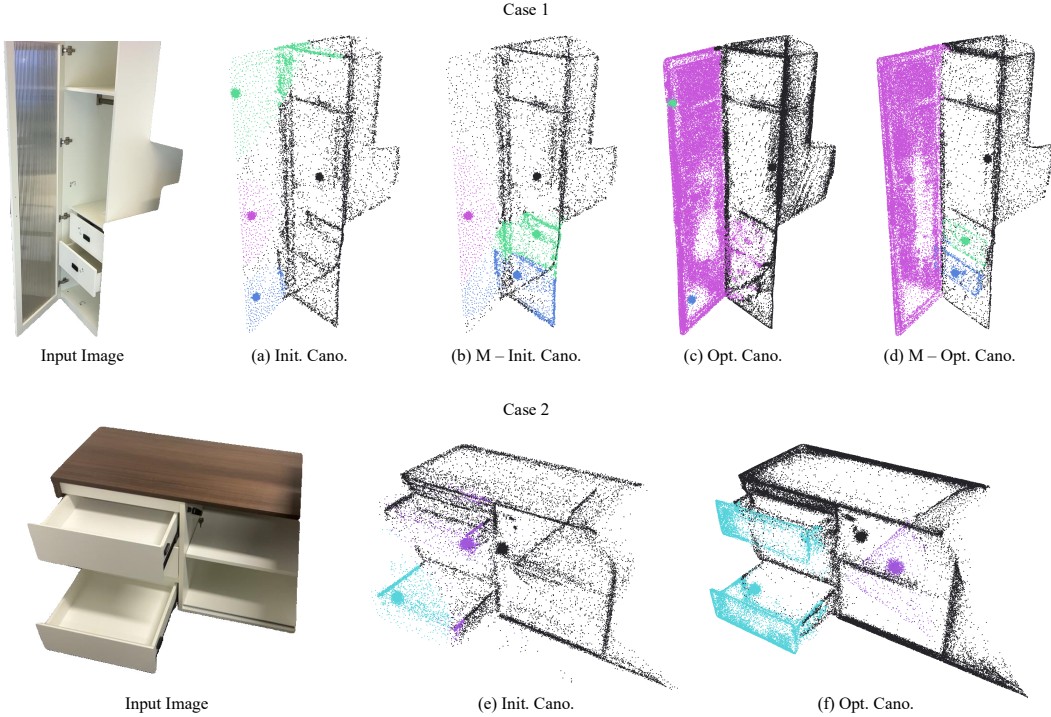

Figure A.1: **Failure cases**. We illustrate failure cases of our ArtGS. 'Init./Opt. Cano.' represents initialized and optimized Canonical Gaussians, while the prefix 'M' indicates manual correction of erroneous part centers.

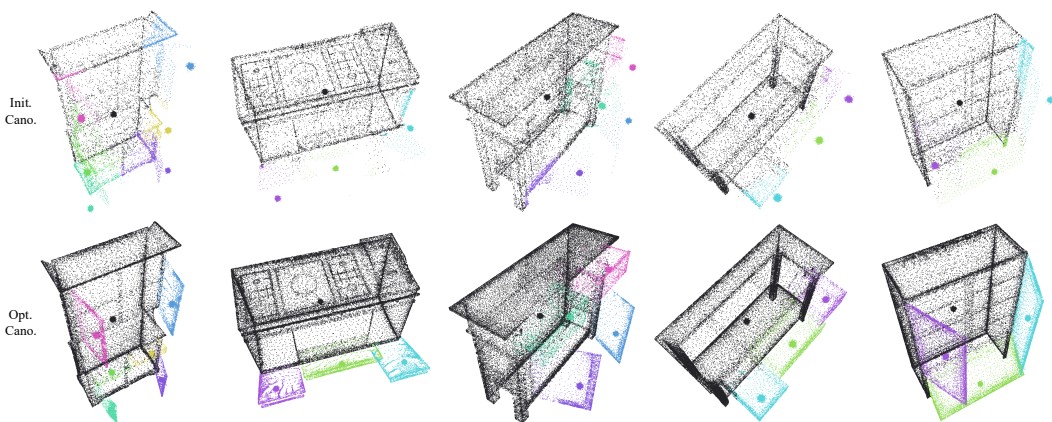

Figure A.2: **Evolution of canonical Gaussians**. We visualize the evolution of canonical Gaussians, showing both their part assignments and centers. Our initialization strategy begins with dense static Gaussians and sparse dynamic Gaussians. As training progresses, the Gaussians undergo densification while simultaneously refining their part centers and assignments. These visualization results demonstrate the effectiveness of ArtGS.

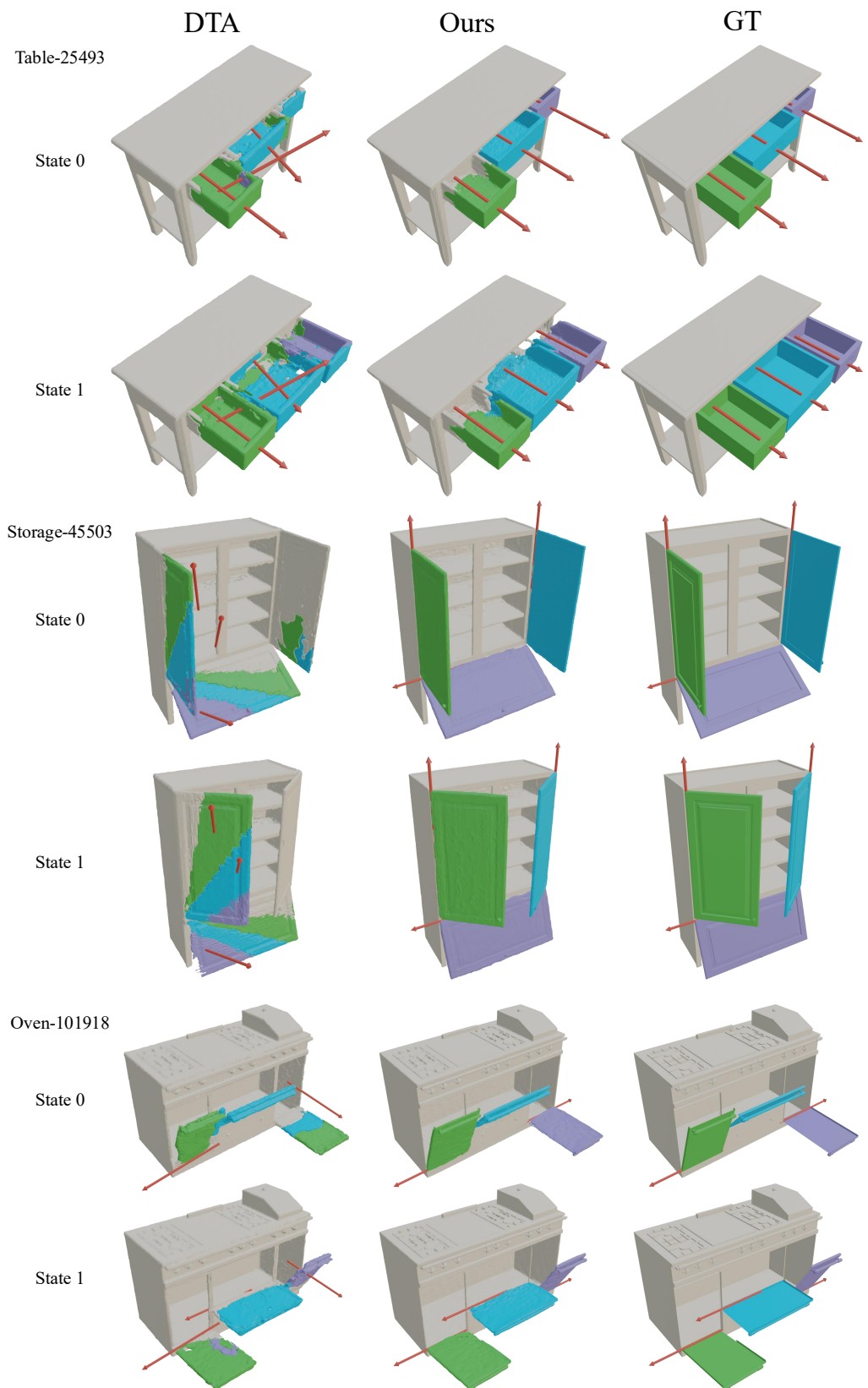

Figure A.3: **Additional qualitative results on ArtGS-Multi.**

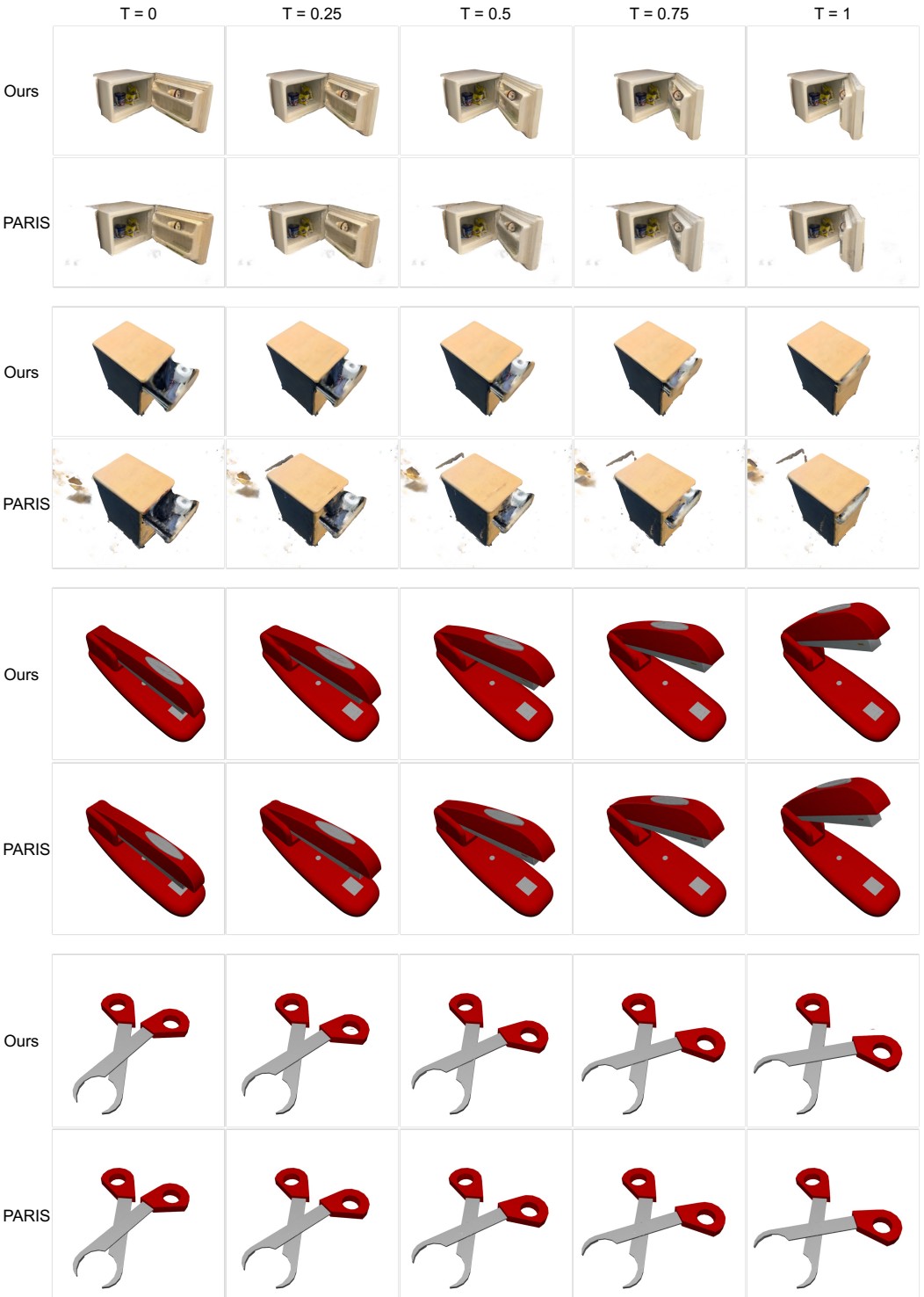

Figure A.4: **Interpolation results on PARIS data.**

