# OpenReview forum: "Building Interactable Replicas of Complex Articulated Objects via Gaussian Splatting"
_ICLR.cc/2025/Conference — ICLR 2025 Poster_

### Official Review · Reviewer_x9y2 · 2024-10-27

**Soundness:** 3
**Presentation:** 2
**Contribution:** 2
**Rating:** 6
**Confidence:** 3

**Summary:**

This paper presents a novel method for interactable articulated object modeling. The method leverages the skinning-like motion priors between object parts and the correspondences between two-state objects to model part-aware articulated objects. Results show that the results outperform existing works in both well-used benchmarks and datasets created by the author containing complex multi-part objects.

**Strengths:**

+ The writing of the paper is easy to follow.
+ The motivation of the paper is reasonable. By combining explicit 3D Gaussian modeling with proper articulation modeling, the method enables better articulation parameter estimation and object reconstruction.
+ The design for initialization and part segmentation based on different states and object parts is reasonable.
+ Extensive experiments are conducted to evaluate the effectiveness of the method presented.

**Weaknesses:**

- The method is designed for two-state objects. It is uncertain how this can be applied to object captures with more states. Moreover, the number of parts is set manually as hyperparameters making it difficult to scale to different objects.
- It seems the method combines existing designs to achieve 3DGS-based articulated modeling. Though it is not super interesting, the overall pipeline makes sense. However, I am not sure what is the argument for using 3DGS compared to NeRF and MVS point clouds.
- The authors only compare the method with DTA for visualization and with DTA and PARIS on DETA-Multi and ArtGS-Multi. Is it possible to compare with more methods? For example, some NeRF-based methods. Some related suggestions:
 [1] Real2Code: Reconstruct Articulated Objects via Code Generation; [2] Articulated Object Neural Radiance Field;
- More visualization results are essential for understanding the method to complement Fig. 4. For example, how well does the matching process and refinement work? I think these visualizations are vital for multi-stage methods.
- I am not sure why perception-based metrics are not reported for evaluations, e.g., PNSR and LPIPS for 3DGS evaluation.
- Though the paper is motivated by interactive articulation objects, I did not see related evaluation and applications for it.
- I did not see discuss about failure cases and limitations.

**Questions:**

- What is the comparison baseline for w/o Cano. init? Does it work if we just take one-state GS as initialization?  I think it should be ok as two-state is only w.r.t. some partial rigid transformations.
- Can you explain the gap between obvious improvement in articulation parameters estimation and minor improvement in overall, geometry modeling?

**Details Of Ethics Concerns:**

Not appliable.

---

> ### Author Response · Authors · 2024-11-22
> **Response to the Reviewer (1 / 3)**
>
> We gratefully appreciate the reviewer for acknowledging our contribution and the effectiveness of our method. Please refer to the general response for a summary of all changes we have made in this revision. We make further clarifications to address the reviewer's concerns as follows:
>
> > Q1. Extension for more states and the number of parts.
> - Extension beyond two states. While our current implementation focuses on two states, our method can be extended to multiple states by (1) using the same canonical Gaussians as a bridge between all states, (2) extending articulation parameters $\Psi$ for multiple states, either optimize a state-depend function that $\Psi_t =f(t)$ or optimize a set of learnable parameters {$\Psi_t $}$_{t=0}^T$ . We leave this for future work.
> - Number of parts. Previous works assumed the number of parts, and we followed their setting. In fact, the number of parts can be easily obtained with GPT-4o. Automating the entire pipeline is worthwhile future work.
>
> > Q2. Advantages of 3D Gaussians.
>
> 3D Gaussians offer several key advantages over NeRF and MVS point clouds for articulated object reconstruction.
> - Compared with NeRF:
> 	- Explicit 3D representation enables direct part assignment and motion modeling, achieving more precise articulation estimation.
> 	- Significantly faster training (3-4x speedup over DTA) and rendering.
> - Compared with point clouds
> 	- We can obtain high-quality images and meshes through 3D Gaussians.
>
> > Q3. Additional baselines for visualization comparison.
>
> We thank the reviewer for suggesting Real2Code and Articulated Object Neural Radiance Field.
> - We tried to compare our ArtGS with Real2Code but we found that it only releases the training code, lacking pre-trained checkpoints and training data. Given that Real2Code is also a submission paper for ICLR 2025, we may not be able to make a comparison before the deadline of rebuttal.
> - [Articulated Object Neural Radiance Field](https://github.com/zubair-irshad/articulated-object-nerf) is the experimental implementation of a few papers including [CLA-NeRF](https://arxiv.org/abs/2202.00181) (ICRA 2022) and [NASAM](https://arxiv.org/abs/2205.08525) (CVPR 2022). We only found the data of laptop-11586 from their [data link 2](https://www.dropbox.com/scl/fi/q88wzo9gk3c1kab7suckx/sapien_single_scene_art.tar.xz?rlkey=ckmpsp8csqm7nz40rfv86w2tv&dl=1) and [data link1](https://www.dropbox.com/scl/fi/t9wsvgnmvjjzpc555nan5/sapien_single_scene.tar.xz?rlkey=mhj1e6jc3ywbvdyl3mwtr70yw&dl=1) is expired. Thus we provide the results of ArtGS on the laptop data in the [anonymous project website](https://artgs-iclr.github.io/artgs-iclr/). The qualitative comparison shows that our method works better.
>
> We added more visual comparisons with PARIS in the [anonymous project website](https://artgs-iclr.github.io/artgs-iclr/) and our paper revision to address the reviewer's concern.

---

> > ### Comment · Reviewer_x9y2 · 2024-11-26
> >
> > Thank you for your replies.
> > The responses address my concerns about initialization and comparisons.
> > However, I am still uncertain about the advantage of 3DGS over point clouds, especially since the authors argue that their major focus is geometry and pose estimation.
> > Considering the overall technical contribution, I maintain my original rating.

---

> > > ### Author Response · Authors · 2024-11-27
> > >
> > > Thank you for your feedback.
> > > We would like to clarify the key advantages of 3D Gaussians over point clouds:
> > >
> > > 1. 3D Gaussians enable the reconstruction of high-quality meshes, which is an ideal representation for creating interactive digital twins in simulation environments.
> > > 2. While point clouds often produce incomplete or low-quality renderings, 3D Gaussians provide high-fidelity rendering capabilities from arbitrary viewpoints. Although our primary focus is on geometry and articulation parameters, our extensive results on the [anonymous project website](https://artgs-iclr.github.io/artgs-iclr/) demonstrate consistently high rendering quality across various synthetic and real-world articulated objects.
> > > 3. As noted in DTA, point cloud reconstruction quality is often limited by sensor noise during data capture. Our 3D Gaussian approach offers enhanced robustness by primarily utilizing multi-view RGB images, significantly reducing the strict depth measurement requirements of point cloud methods.
> > >
> > > | Method | FoldChair | Fridge | Laptop | Oven | Scissor | Stapler | USB | Washer | Blade | Storage | All | Fridge | Storage | All |
> > > |:--:|:--:|:--:|:--:|:--:|:--:|:--:|:--:|:--:|:--:|:--:|:--:|:--:|:--:|:--:|
> > > | DTA | **0.26** | 0.70 | **0.34** | 4.25 | **0.41** | **2.02** | **1.34** | **4.53** | **0.37** | **4.04** | **1.83** | 2.13 | 9.01 | 5.57 |
> > > | TSDF with gt depth | 0.30 | **0.56** | 0.47 | **3.60** | 0.49 | 2.78 | 1.60 | 5.73 | 0.54 | 5.13 | 2.12 | 3.15 | 131.86 | 67.51 |
> > > | Ours | 0.36 | 0.59 | 0.48 | 3.64 | 0.68 | 2.88 | 1.58 | 6.05 | 0.63 | 5.17 | 2.21 | **1.37** | **2.84** | **2.11** |
> > >
> > > As shown in this table, for real-world objects, TSDF with ground truth depth on real-world objects may produce poor-quality meshes (e.g., real\_storage) due to depth sensor noise, while ArtGS achieves high-quality reconstruction.
> > >
> > > We hope our response has thoroughly addressed your concerns. We are open to further discussion regarding any remaining questions or concerns. If our response and additional results have addressed your concerns, we would greatly appreciate your consideration of a higher score. Your suggestions are instrumental in improving the quality of our paper, and we sincerely thank you for providing your valuable feedback.

---

> ### Author Response · Authors · 2024-11-22
> **Response to the Reviewer (2 / 3)**
>
> > Q4. More visualization results for each stage.
>
> We will provide more visualizations of the initialized canonical Gaussians and optimized Gaussians in the appendix.
>
> > Q5. Perception-based metrics.
>
> Following DTA, we focus on the quality of mesh reconstruction and articulation estimation, thus we do not report perception-based metrics. Since DTA doesn't provide code to measure perception-based metrics, we report the results of PARIS and ArtGS in Tab.7. As shown in Tab.7, ArtGS achieves comparable or superior performance relative to PARIS. We also show results here:
>
> | Metric | Method | FoldChair | Fridge | Laptop | Oven | Scissor | Stapler | USB | Washer | Blade | Storage | All | Fridge | Storage | All |
> |:--:|:--:|:--:|:--:|:--:|:--:|:--:|:--:|:--:|:--:|:--:|:--:|:--:|:--:|:--:|:--:|
> | PSNR | PARIS | 31.50 | **37.67** | **37.26** | 35.30 | **38.37** | 38.49 | 39.07 | **40.08** | 38.29 | 36.18 | 37.22 | 25.29 | **27.13** | 26.21 |
> | | Ours | **34.46** | 37.11 | 34.09 | **37.06** | 38.29 | **39.13** | **39.64** | 38.50 | **41.16** | **37.24** | **37.67** | **27.05** | 25.38 | **26.22** |
> | SSIM | PARIS | 0.985 | **0.994** | **0.991** | 0.980 | 0.996 | 0.995 | 0.992 | 0.991 | 0.996 | **0.993** | 0.991 | 0.898 | **0.953** | 0.926 |
> | | Ours | **0.997** | 0.993 | 0.988 | **0.995** | **0.998** | **0.999** | **0.998** | **0.995** | **0.999** | 0.992 | **0.995** | **0.939** | 0.930 | **0.935** |
> | LPIPS_vgg | PARIS | 0.045 | **0.032** | **0.020** | **0.045** | 0.015 | 0.019 | 0.029 | **0.029** | 0.017 | **0.095** | **0.035** | 0.188 | **0.139** | 0.164 |
> | | Ours | **0.036** | 0.041 | 0.045 | 0.054 | **0.014** | **0.011** | **0.016** | 0.052 | **0.004** | 0.097 | 0.037 | **0.114** | 0.188 | **0.151** |
>
> > Q6. Applications of ArtGS.
>
> The interactable meshes reconstructed by ArtGS can be put into the simulator to train embodied robots. This is an immediate application but not the problem our paper aims to address. We provide a video in the [anonymous project website](https://artgs-iclr.github.io/artgs-iclr/) for simulation results in IsaacSim.
>
> > Q7. Failure cases and limitations
> 1. Failure cases.
> We will add a visualization of failure cases in the appendix soon and we describe them here.
> - For real-world objects with multiple parts, some part centers derived from clustering may be inaccurate, and these incorrectly initialized centers may not be corrected during the optimization process, resulting in degraded performance for parts with misaligned centers. However, we found that if we manually correct the wrong part centers before training (i.e. modifying the position of the wrong part centers), it works well.
> - Similar motion. If two parts move in the same way in two states, such as two drawers being pulled apart together, our method may fail.
> 2. Limitations. We've discussed some limitations of our method in Sec.C in the supplementary, and we briefly summarize it here. Please refer to Sec.C for detailed discussions.
> - Stability of randomness. The stability issues often stem from the initialization of three key components: canonical Gaussians $\mathcal{G}^c$, part centers $C$, and joint articulation parameters $\Psi$. While our initialization strategies have greatly improved stability, severe initialization errors in $\mathcal{G}^c$ and $C$ may result in part mis-segmentation. Integrating prior models such as SAM will help to enhance the ability to correct center errors.
> - Limited states. Our approach is limited to two states, which may not fully capture the complexity of real-world objects. Extending our method to more states or videos is worthwhile future work.
> - Mesh reconstruction fidelity. We use the original Gaussian Splatting as implementation, which is limited in mesh reconstruction compared with NeRF-based methods. Integrating recent advancements in reconstruction with Gaussian Splatting may be helpful.

---

> > ### Author Response · Authors · 2024-11-25
> > **Visualization of canonical Gaussians and Failure Cases**
> >
> > In the latest revision, we add **a visualization of failure cases** (Fig. 5) and **a visualization of canonical Gaussian evolution** (Fig. 6) in the appendix.
> > > Failure Cases.
> >
> > **Case.1 Incorrect Initialization of Part Centers.** For real-world objects with multiple parts, clustering-derived part centers may be inaccurate (Fig. 5 (a)) due to sensor noise, occlusion, and varying illumination conditions. These incorrectly initialized centers often persist through optimization, degrading performance for parts with misaligned centers (Fig. 5 (c)). Manual correction of erroneous part centers prior to training (Fig. 5 (b)) yields improved results (Fig. 5 (d)). As discussed in Appendix C, incorporating prior models like SAM for automatic, accurate part center initialization remains a promising direction for future work.
> >
> > **Cases.2 Similar Motions.** Our method exhibits limitations when handling parts with identical motion across states, as demonstrated in case 2 of Fig.5 where two drawers are pulled with the same distance. In such scenarios, the model tends to learn a single joint to fit both parts, failing to distinguish between the independently movable parts. As discussed in Sec.C, expanding ArtGS to incorporate additional states would provide richer motion information, potentially enabling better part separation.
> >
> > > Canonical Gaussian Evolution
> >
> > We visualize the evolution of canonical Gaussians in Fig.6, showing both their part assignments and centers. Our initialization strategy begins with dense static Gaussians and sparse dynamic Gaussians. As training progresses, the Gaussians undergo densification while simultaneously refining their part centers and assignments. These visualization results demonstrate the effectiveness of ArtGS.
> >
> > Note: Due to the addition of new visualizations, the interpolation results previously referenced as Fig. 6 in the latest revision are now shown in Fig. 8.
> >
> > We welcome further discussion regarding any remaining concerns.

---

> ### Author Response · Authors · 2024-11-22
> **Response to the Reviewer (3 / 3)**
>
> > Q8. Canonical Gaussian initialization.
>
> For "w/o Cano. init", we randomly initialize canonical Gaussians.
> - **Using single-state Gaussians $\mathcal{G}_{single}^{0/1}$ to initialize the canonical Gaussians.** As discussed in Sec.4, we set the canonical state as $t = 0.5$. We found in experiments that using single-state Gaussians to initialize the canonical Gaussians easily converges to local optimal.
> Assuming we use $\mathcal{G}_{single}^0$ as initialization, then the loss of state 0 will be very small, while the loss of state 1 will be very large. This asymmetry will cause the model to fail to correct the canonical Gaussian to the ideal t=0.5. In the end, most Gaussians remain static, leading to incorrect articulation parameters.
> - If we set the canonical state as t=0/1, lacking the motion consistency that  $T^{c\rightarrow0}=(T^{c\rightarrow1})^{-1}$, the model will easily collapse or degenerate, especially on complex objects. In this setting, a local optimal found by the model is that all Gaussians have low opacity, which prevents the loss of another state from being too large.  However, since the pruning strategy will eliminate the Gaussians with low opacity, all Gaussians are pruned and it results in training collapse.  We tried multiple sets of hyper-parameters, but the performance was extremely erratic.
>
> > Q9. Obvious improvements in articulation parameters estimation.
>
> Our ArtGS demonstrates obvious improvements in articulation parameters estimation due to:
> 1. Explicit 3D representation of GS enables direct part assignment and motion modeling, achieving more precise articulation estimation.
> 2. Our center-based part assignment module utilizes spatial information, which brings an enhanced decomposition capability.
> 3. Our novel initialization strategies, including canonical Gaussian and part centers, greatly reduce the difficulty of training the model and helping it converge to a better solution.
>
> > Q10. Minor improvement in geometry reconstruction.
>
> For geometry reconstruction, ArtGS demonstrates large improvement on complex or real-world objects but has minor improvement on simple synthetic objects. This results from two reasons:
> 1. TSDF limitation. Our method's performance on simple synthetic objects, particularly in terms of CD-w metric, is constrained by our use of TSDF for mesh extraction from Gaussian Splatting-rendered depths. To analyze this limitation, we compare against meshes reconstructed using ground-truth depth with TSDF.
>
> We report the results here:
>
> | Method | FoldChair | Fridge | Laptop | Oven | Scissor | Stapler | USB | Washer | Blade | Storage | All | Fridge | Storage | All |
> |:--:|:--:|:--:|:--:|:--:|:--:|:--:|:--:|:--:|:--:|:--:|:--:|:--:|:--:|:--:|
> | DTA | **0.26** | 0.70 | **0.34** | 4.25 | **0.41** | **2.02** | **1.34** | **4.53** | **0.37** | **4.04** | **1.83** | 2.13 | 9.01 | 5.57 |
> | TSDF with gt depth | 0.30 | **0.56** | 0.47 | **3.60** | 0.49 | 2.78 | 1.60 | 5.73 | 0.54 | 5.13 | 2.12 | 3.15 | 131.86 | 67.51 |
> | Ours | 0.36 | 0.59 | 0.48 | 3.64 | 0.68 | 2.88 | 1.58 | 6.05 | 0.63 | 5.17 | 2.21 | **1.37** | **2.84** | **2.11** |
>
> As shown in this table, even with ground-truth depth input, TSDF-based reconstruction cannot surpass algorithms using marching cubes with NeRF, primarily due to the fundamental differences between TSDF and marching cubes algorithms on simple geometries. However, for complex or real-world objects where articulation reconstruction becomes more critical, the advantages of our model become evident. Additionally, TSDF with ground truth depth on real-world objects may produce poor-quality meshes (e.g., real\_storage) due to depth sensor noise, while ArtGS achieves high-quality reconstruction.
>
> 2. As discussed in Sec.C, our current implementation adopts the original Gaussian Splatting, which has limited quality for mesh reconstruction  compared with NeRF-based methods. Future work could enhance mesh reconstruction quality by incorporating recent advances in Gaussian Splatting (e.g. 2D Gaussians).
>
> Importantly, our primary objective is to create digital twins of real-world articulated objects, where ArtGS demonstrates significant performance improvements.
>
> We hope the above response can resolve your questions and concerns. Please let us know if there is any further questions!

---

### Official Review · Reviewer_AX3x · 2024-10-30

**Soundness:** 2
**Presentation:** 2
**Contribution:** 3
**Rating:** 6
**Confidence:** 4

**Summary:**

- This paper works on the task of articulated object reconstruction from multi-view RGB-D images.
- This work proposes to leverage 3D Gaussian as the underline representation to reconstruct articulated objects with multiple parts in a self-supervised way.
- Along with the representation, several strategies are proposed to improve the reconstruction accuracy, such as a coarse-to-fine initialization for part geometry and skinning-inspired dynamics modeling for articulation estimation.
- Experiments demonstrate the effectiveness and robustness of the proposed method.

**Strengths:**

-	This work is contributing to an increasingly important research problem of building digital twins for arbitrary articulated objects with complex structure.
-	The experiments are showing promising results overall.
-	The observation and analysis of the experiment results are well discussed.
-	This work is well contextualized in the first two sections.

**Weaknesses:**

-	**The writing of the Method section should be improved**. The quality of the writing overall is great, but the method section requires further improvement to articulate the details of the approach, especially for the logic of the presentation and usage of mathematical symbols. The current form hinders the understanding of the proposed method. Please see more details for clarification in the `Question` section.
-	**The experiment evaluation needs a bit more care**.
	-	There are at least 17 highlighting of numbers in `Table 1` is misplaced, e.g., PARIS*[Axis Ang, Real Objects-Fridge] in the second row should be the second best.
	-	The “Axis Pos” reported in `Table 5` for synthetic objects is almost no difference given the limited significant figures. It would be more meaningful to show the metrics with more significant figures or report the numbers by timing 10 or even 100.
-	**More qualitative results**. The presentation of the experiment would benefit from showing more qualitative results (in different complexity) that compares with the baseline methods, which can visually demonstrate the gap revealed in the metric evaluation.
-	**More discussion of the failure cases and limitations**. For the completeness of the paper, the failure cases and discussion of the limitation accordingly should be incorporated. There are assumptions that are not explicitly mentioned in the paper, e.g., the number of parts is given, the two states of the observation are pre-aligned in the world/object coordinate system. Maybe there are other assumptions that can be discussed, e.g., whether the articulated parts are required to be opened to expose visibility as much as possible.

**Questions:**

The below are some details of the approach or implementation that would benefit from clarification.

-	In Section 4.1:
	-	Once $ \mathcal{G}_{init}^c$ is initialized, are the Gaussians labeled as static/dynamics in binary or part ids?
	-	Since the refinement of $ \mathcal{G}_{coarse}^c$ only involves the static part, would the region of the movable part be very sparse? Does this sparsity remain for the optimization step later? Would it affect the reconstruction quality in the end?
	-	How many Gaussians are initialized in $ \mathcal{G}_{init}^c$?
-	In Section 4.2:
	-	What are the learnable centers $C_k$ useful for by intuition? What is $\mathbf{R}_k$ in $C_k$? Is $\mathbf{R}_k$ related to the articulation for the specific part?
	-	In equation 6, the variables $X_i^k$,  $\mu_i^c$, and $\mathbf{W}_\Delta$ are not defined.
	-	In lines 256-257, it is unclear how the distance matrix is related to part assignment. It is also unclear why identifying the sharp boundaries is the unique challenge in this case.
	-	In lines 258-260, why introducing $\mathbf{W}_\Delta$ is helpful in tackle the challenge?
-	In Section 4.3:
	-	In equation 7, $M_{ik}$ is undefined.
	-	In equation 8, $\mathbf{r}_i^c$ is undefined.

The below are some missing references for citation.
- [1] Yan, Zihao, et al. "RPM-Net: recurrent prediction of motion and parts from point cloud." arXiv preprint arXiv:2006.14865 (2020).
- [2] Jain, Ajinkya, et al. "Screwnet: Category-independent articulation model estimation from depth images using screw theory." 2021 IEEE International Conference on Robotics and Automation (ICRA). IEEE, 2021.
- [3] Lei, Jiahui, et al. "Nap: Neural 3d articulated object prior." Advances in Neural Information Processing Systems 36 (2023): 31878-31894.
- [4] Liu, Jiayi, Manolis Savva, and Ali Mahdavi-Amiri. "Survey on Modeling of Human-made Articulated Objects." arXiv preprint arXiv:2403.14937 (2024).

---

> ### Author Response · Authors · 2024-11-22
> **Response to the Reviewer (1 / 2)**
>
> We gratefully appreciate the reviewer for acknowledging our contribution and the effectiveness of our method. We also thank the reviewer for constructive and detailed feedback. Please refer to the general response for a summary of all changes we have made in this revision. We make further clarifications to address the reviewer's concerns as follows:
>
> > Q1. Questions for the method section.
> - Section 4.1
>     - Label static/dynamic Gaussians. We distinguish static/dynamic Gaussians only for initializing canonical Gaussians and part centers. Once $\mathcal{G}_{init} ^c$ and part centers are initialized, we no longer distinguish static/dynamic Gaussians by motion prior. Instead, we optimize the part assignment module to obtain the part IDs of each Gaussian. We set part 0 as the static root part.
>     - Sparsity of movable Gaussians. The obtained  $\mathcal{G}_{init} ^c$ has dense static Gaussians and sparse dynamic Gaussians. We densify all Gaussians during the training process. In the end, we obtain dense static and dynamic Gaussians, achieving high-quality reconstruction.
>     - The number of initialized Gaussians. About 10k-50k Gaussians will be initialized in  $\mathcal{G}_{init} ^c$ , depending on the specific object.
> - Section 4.2
>     - Intuition of learnable centers and  $R_{k}$'s relation to articulation. Because we do not know the centers of each part, we model them as learnable parameters and optimize these learnable centers for better part assignments.  $R_{k}$  is not related to the articulation.
>     - Definition of $X_{i}^{k},\mu_{i}^{c},W_\Delta$. $X_{i}^{k}$  is the relative position of Gaussian $i$ in the coordinate of part $k$, $\mu_{i}^{c}$ is the position of Gaussian $i$ in the canonical space, and $W_\Delta=\mathrm{MLP}(\mu,X,D)$ is a residual term described at line 258-260.
> - Sec 4.3
>     - Definition of $M_{ik}$. $M$ has been defined in Eq.6, and $M_{ik}$ denotes the probability of that Gaussian $i$ is belong to part $k$ .
>     - Definition of $r_{i}^c$. We use $r_{i}$ to denote the rotation of Gaussian $i$ (line 144-145), and $r_{i}^c$ means the rotation of Gaussian $i$ in the canonical space.
>
> > Q2. Experiments evaluation.
> - Highlighting problems. Thanks a lot for such detailed feedback. We've corrected all highlighting misplacements in our revision.
> - Axis Pos metric timed by 1000. Following DTA and PARIS, we multiply the 'Axis Pos' metric by 10 in Tab.1 and Tab.5. We report the 'Axis Pos' metric multiplied by 1000 in Tab.6 and ArtGS demonstrates superior performance compared to DTA. We also show the results here:
>
> | Metric | Method | FoldChair | Fridge | Laptop | Oven | Scissor | Stapler | USB | Washer | All |
> |:--:|:--:|:--:|:--:|:--:|:--:|:--:|:--:|:--:|:--:|:--:|
> | Axis Pos | DTA | 0.53±0.3 | 0.62±0.3 | 1.10±0.7 | 1.49±1.0 | 2.48±2.8 | 2.21±1.8 | 0.35±0.2 | 4.53±2.8 | 1.66±1.2 |
> | | Ours | **0.48±0.2** | **0.44±0.2** | **0.39±0.3** | **0.55±0.4** | **0.16±0.1** | **0.93±0.4** | **0.08±0.1** | **0.33±0.3** | **0.42±0.3**|

---

> ### Author Response · Authors · 2024-11-22
> **Response to the Reviewer (2 / 2)**
>
> > Q3. More qualitative results.
>
> We have added comprehensive video results and interactable meshes in the [anonymous project website](https://artgs-iclr.github.io/artgs-iclr/). These videos clearly demonstrate our method's advantages in handling complex dynamic scenarios and its practical utility for creating interactive digital twins.
>
> > Q4. Failure cases and limitations.
> 1. Limitations. We've discussed some limitations of our method in Sec.C in the supplementary, and we briefly summarize it here. Please refer to Sec.C for detailed discussions.
> - Stability of randomness. The stability issues often stem from the initialization of three key components: canonical Gaussians $\mathcal{G}^c$, part centers $C$, and joint articulation parameters $\Psi$. While our initialization strategies have greatly improved stability, severe initialization errors in $\mathcal{G}^c$ and $C$ may result in part mis-segmentation. Integrating prior models such as SAM will help to enhance the ability to correct center errors.
> - Limited states. Our approach is limited to two states, which may not fully capture the complexity of real-world objects. We will extend our method to more states or videos in future work.
> - Mesh reconstruction fidelity. We use the original Gaussian Splatting as implementation, which is limited in mesh reconstruction compared with NeRF-based methods. Integrating recent advancements in reconstruction with Gaussian Splatting may be helpful.
> - Number of parts. Previous works assumed the number of parts, and we followed their setting. In fact, the number of parts can be easily obtained with GPT-4o. Automating the entire pipeline is worthwhile future work.
> - Pre-aligned states. All we need is to make all cameras in both states to be aligned. Synthetic data has pre-aligned cameras. For real-world data, we can use either ICP alignment or continuous video capture to maintain consistent camera coordinates.
> - Object state. More visibility might lead to better reconstruction. But we don't need parts to be opened to expose visibility as much as possible, we just need the parts to have different states in the two states, providing enough motion information to identify these movable parts.
> 2. Failure cases.
>
> We will add a visualization of failure cases in the appendix soon and we describe them here.
> - For real-world objects with multiple parts, some part centers derived from clustering may be inaccurate, and these incorrectly initialized centers may not be corrected during the optimization process, resulting in degraded performance for parts with misaligned centers. However, we found that if we manually correct the wrong part centers before training (i.e. modifying the position of the wrong part centers), it works well.
> - Similar motion. If two parts move in the same way in two states, such as two drawers being pulled apart together, our method may fail.
>
> > Q5. Missed References.
>
> We have added these references in our revision.
>
> We hope the above response can resolve your questions and concerns. Please let us know if there is any further question!

---

> > ### Author Response · Authors · 2024-11-25
> > **Visualization of Failure Cases**
> >
> > In the latest revision, we add **a visualization of failure cases** (Fig. 5) in the appendix.
> >
> > **Case.1 Incorrect Initialization of Part Centers.** For real-world objects with multiple parts, clustering-derived part centers may be inaccurate (Fig. 5 (a)) due to sensor noise, occlusion, and varying illumination conditions. These incorrectly initialized centers often persist through optimization, degrading performance for parts with misaligned centers (Fig. 5 (c)). Manual correction of erroneous part centers prior to training (Fig. 5 (b)) yields improved results (Fig. 5 (d)). As discussed in Appendix C, incorporating prior models like SAM for automatic, accurate part center initialization remains a promising direction for future work.
> >
> > **Cases.2 Similar Motions.** Our method exhibits limitations when handling parts with identical motion across states, as demonstrated in case 2 of Fig.5 where two drawers are pulled with the same distance. In such scenarios, the model tends to learn a single joint to fit both parts, failing to distinguish between the independently movable parts. As discussed in Sec.C, expanding ArtGS to incorporate additional states would provide richer motion information, potentially enabling better part separation.
> >
> > Note: Due to the addition of new visualizations, the interpolation results previously referenced as Fig. 6 in the latest revision are now shown in Fig. 8.
> >
> > We welcome further discussion regarding any remaining concerns.

---

> > > ### Comment · Reviewer_AX3x · 2024-11-27
> > >
> > > Thank the authors for the detailed responses.
> > > They addressed most of my concerns and questions, so I will keep my original positive rating.
> > >
> > > To ensure clarity and to meet the writing standards, I hope the authors can further improve writing of the method section for the camera-ready revision. Currently, I haven't seen any changes to the manuscript yet.
> > > - From the explanation above, there are situations in which the math symbol is reused for different purposes, e.g., the $R$ in line 147 and the one in line 292 means different things.
> > > - Please ensure the variables are defined/explained in the text before/after the equations. For example, in line 289, it is better to add "$M_{ik}$ denotes the probability of the Gaussian $i$ belongs to part $k$" in line 289 after "where ...". The definition of $M$ in equation 6 does not mean that $M_{ik}$ is defined.

---

### Official Review · Reviewer_HLUC · 2024-11-03

**Soundness:** 2
**Presentation:** 2
**Contribution:** 2
**Rating:** 6
**Confidence:** 4

**Summary:**

This paper presents ArtGS, an innovative method for reconstructing multi-part interactive objects. ArtGS utilizes 3D Gaussian Splatting combined with a coarse-to-fine standard Gaussian initialization strategy to achieve high-precision reconstruction of dynamic scenes.

**Strengths:**

The key highlight of the paper is the innovative introduction of 3D Gaussian Splatting (GS) and superpoints, along with a coarse-to-fine Gaussian initialization strategy. Additionally, the method achieves state-of-the-art results on existing datasets, particularly excelling on a newly introduced dataset.

**Weaknesses:**

The final visual renderings demonstrated in the paper primarily focus on cabinet-like objects, lacking validation on other categories. To better assess the generalizability of the method, I suggest adding renderings for a broader range of objects, such as chairs, scissors, and staplers, which have distinct motion components and structural differences. This would provide a more comprehensive evaluation of the method's applicability and accuracy across diverse objects.

Moreover, I have some doubts regarding the main contributions and novelty of this work. The success of this method may largely derive from elements such as the coarse-to-fine training strategy, joint optimization of static shape and articulation, the use of Gaussians instead of NeRF, and super-part rigging—all of which have been explored in previous research but are integrated here for this specific task. I would appreciate if the authors could further clarify their unique contributions.

Lastly, there is a lack of video examples to visually demonstrate the interaction of dynamic parts. Including video results would significantly help in understanding the method's advantages in complex dynamic scenarios.

**Questions:**

Please refer to the weaknesses section for further questions.

---

> ### Author Response · Authors · 2024-11-22
> **Response to the Reviewer**
>
> We thank the reviewer for the clear summary of our proposed method and for acknowledging its effectiveness. However, we respectfully disagree with the reviewer simply summarizing our paper as an "integration of explored components". Our contributions extend well beyond mere integration, as recognized by multiple reviewers, including significant improvements on an important research problem (JGzH), reasonable designs for part/articulation modeling and innovative initialization strategies (x9y2), and the effectiveness of our proposed method (JGzH, AX3x, x9y2). Please refer to the general response for a summary of all changes we have made in this revision. We make further clarifications to address the reviewer's concerns as follows:
>
> > Q1. Technical novelty and contributions.
>
> The application of Gaussian Splatting to articulated object reconstruction is far from straightforward, as evidenced by our ablation studies. As shown in Tab.4 and Fig.4, without our proposed initialization strategies, simply integrating Gaussian Splatting and center-based part assinment module together gets very poor results. Our proposed novel techniques try to solve a fundamental challenge: the simultaneous optimization of canonical Gaussians, part assignment, and part articulation. This joint optimization is extremely challenging because errors in any one component can cascade and cause the entire system to fail. Our method systematically addresses this challenge through:
> - Better Canonical Gaussian Design:
>     - Set the canonical Gaussian state as t=0.5 for natural motion consistency (i.e. $T^{c\rightarrow0}=(T^{c\rightarrow1})^{-1}$), which is significant for preventing the collapse of optimization.
>     - A novel matching-based initialization strategy that provides better starting points for optimization, and helps the model find better solutions.
>     - Leverage motion information to identify static and dynamic parts.
> - Spatial-aware Part Discovery with Motion Information.
>     - Introduce a center-based part assignment module that leverages the spatial and dynamic information of 3D GS.
>     - Utilize motion information and clustering for part center initialization.
>
> These novel designs effectively solve the challenge and bring insight that providing good initialization of canonical Gaussians, part assignment, and part articulation is essential for articulated object reconstruction.
>
> > Q2. Visual renderings of more categories.
>
> While our paper visualizes representative examples of multi-part objects, our method successfully handles diverse object categories. We have added comprehensive visual comparisons in the [anonymous project website](https://artgs-iclr.github.io/artgs-iclr/), showing successful reconstructions across all these categories. These results demonstrate our method's strong generalizability across different motion types and structural complexities.
>
> > Q3. More visualization results for video.
>
> We have added comprehensive video results and interactable meshes in the [anonymous project website](https://artgs-iclr.github.io/artgs-iclr/). These videos clearly demonstrate our method's advantages in handling complex dynamic scenarios and its practical utility for creating interactive digital twins.
>
> We hope the above response can resolve your questions and concerns. Please let us know if there is any further question!

---

> ### Author Response · Authors · 2024-11-27
>
> Dear reviewer,
>
> We hope our response has addressed your questions and concerns. If so, could we kindly ask you to consider increasing the score? Thank you once again for your valuable feedback!

---

### Official Review · Reviewer_JGzH · 2024-11-04

**Soundness:** 4
**Presentation:** 3
**Contribution:** 3
**Rating:** 6
**Confidence:** 4

**Summary:**

This paper introduces a part-aware articulated object reconstruction method with 3D Gaussian splatting. The method reconstructs the shape and articulation from two states of multi-view RGB-D images with a coarse-to-fine Gaussian initialization. A skinning-related center-based part modeling is designed to automatically assign Gaussians to statics and movable parts. The articulation type is learned through self-guided parameter learning.

**Strengths:**

1. The paper is well-organized and the overall writing is good.

2. The optimization speed is improved by using 3D Gaussian splitting.

3. Center-based part modeling and assignment allow the model to assign Gaussian to different movable and static parts. The use of part centers allows multi-movable parts compared to PARIS and does not require prior part segmentation knowledge compared to DTA.

4. The self-guided motion priors with part dual-quaternions improve articulation learning with a further heuristic joint type prediction.

5. The experiments and ablation studies are extensive.

**Weaknesses:**

1. Only state 0 and state 1 results are presented, the authors should present more intermediate states like Figure 8 in PARIS where they illustrated results on t=0.25, t=0.5,  and t=0.75.

**Questions:**

1. Have you tried to present the results when different movable parts are in different states?  Since this method models multiple movable parts with distinct part centers and dual quaternions, I believe it is possible to do that. It would be interesting to see those results.

---

> ### Author Response · Authors · 2024-11-22
> **Response to the Reviewer**
>
> We gratefully thank the reviewer for acknowledging the effectiveness and efficiency of our proposed model. Please refer to the general response for a summary of all changes we have made in this revision. We make further clarifications to address the reviewer's concerns as follows:
> > Q1. Intermediate states visualization.
>
> We agree that showing intermediate states would better demonstrate our method's capabilities. We have added Fig.6 in the supplementary, showing rendering results at t=0, 0.25, 0.5, 0.75, and 1 for various objects. We also render a continuous video in the [anonymous project website](https://artgs-iclr.github.io/artgs-iclr). Our method produces smooth and physically plausible interpolations between states across all objects.
>
> > Q2. Present the results when different parts are in different states.
>
> Our method can indeed handle different movable parts in different states. Actually, all movable parts are in two different states, which provide motion clues for decomposing them. This flexibility stems from our part-aware design where each part's transformation is independently optimized through its own dual-quaternion parameters, allowing for arbitrary combinations of part states.
>
> We hope the above response can resolve your questions and concerns. Please let us know if there is any further question!

---

> > ### Comment · Reviewer_JGzH · 2024-11-26
> >
> > Thank the authors for the response and additional results. I will keep my positive rating.

---

> > > ### Author Response · Authors · 2024-11-27
> > >
> > > Thanks for your reply. We hope our response has thoroughly addressed your concerns. We are open to further discussion regarding any remaining questions or concerns. If our response and additional results have addressed your concerns, we would greatly appreciate your consideration of a higher score. Your suggestions are instrumental in improving the quality of our paper, and we sincerely thank you for providing your valuable feedback.

---

### Author Response · Authors · 2024-11-22
**General response to the reviewers**

We sincerely thank reviewers JGzH, HLUC, AX3x, x9y2 for their valuable feedback, especially highlighting our contributions on reasonable motivation (x9y2), extensive experiments and analyses (JGzH, AX3x, x9y2), effective designs for canonical Gaussians and part assignment module (JGzH, x9y2), and innovative initialization strategies (HLUC, x9y2). We follow comments and suggestions from all reviewers and revise our manuscript (colored in blue) accordingly as follows:

1. **More Visualization Results**. (Reviewer JGzH, HLUC, AX3x, x9y2)
- We have created a [anonymous project website](https://artgs-iclr.github.io/artgs-iclr/) with interactive demonstrations and additional results:
    - Simulation video of reconstructed meshes in IsaacSim.
	- Interactable meshes of complex synthetic and real-world objects.
	- Rendering results of all objects at interpolated states.
	- Qualitative comparison of rendering results on various objects at interpolated states and novel views.
	- Please note that as requested by the reviewers, we have included numerous visualization results, which may require some loading time.
- We will provide more visualizations of the initialized canonical Gaussians and optimized Gaussians in the appendix. (Reviewer x9y2)


2. **Additional Experiments and Analysis**.
- We provide the values of Axis Pos multiplied by 1000 in Tab.6 in the appendix. (Reviewer AX3x)
- We added perception-based metrics (PSNR, LPIPS, SSIM) in Tab.7 in the appendix. (Reviewer x9y2)

3. **Failure Cases**. (Reviewer AX3x, x9y2)

We will add a visualization of failure cases in the appendix soon and we describe them here.
- For real-world objects with multiple parts, some part centers derived from clustering may be inaccurate, and these incorrectly initialized centers may not be corrected during the optimization process, resulting in degraded performance for parts with misaligned centers. However, we found that if we manually correct the wrong part centers before training (i.e. modifying the position of the wrong part centers), it works well.
- Similar motion. If two parts move in the same way in two states, such as two drawers being pulled apart together, our method may fail.

4. **Contributions and Novelty Clarification**. (Reviewer HLUC)
- Our key innovation lies in solving the challenging problem of simultaneous optimization of canonical Gaussians, part assignment, and articulation parameters. Our method's success stems from:
    - Canonical Gaussians that effectively bridge 2 object states.
    - A spatial-aware part assignment module that enables accurate unsupervised part-segmentation.
    - Novel initialization strategies for canonical Gaussians and part centers that prevent optimization collapse.
- As acknowledged by all reviewers, with all the innovations proposed in this paper, ArtGS achieves:
    - State-of-the-art performance across diverse object categories.
    - 3-4x faster training compared to the previous SOTA method.
    - No requirement for priors from pre-trained models.
    - handling complex multi-part objects (up to 7 parts).

---

> ### Author Response · Authors · 2024-11-25
> **Additional Visualization Results**
>
> In the latest revision, we add **a visualization of failure cases** (Fig. 5) and **a visualization of canonical Gaussian evolution** (Fig. 6) in the appendix.
> > Failure Cases.
>
> **Case.1 Incorrect Initialization of Part Centers.** For real-world objects with multiple parts, clustering-derived part centers may be inaccurate (Fig. 5 (a)) due to sensor noise, occlusion, and varying illumination conditions. These incorrectly initialized centers often persist through optimization, degrading performance for parts with misaligned centers (Fig. 5 (c)). Manual correction of erroneous part centers prior to training (Fig. 5 (b)) yields improved results (Fig. 5 (d)). As discussed in Sec.C, incorporating prior models like SAM for automatic, accurate part center initialization remains a promising direction for future work.
>
> **Cases.2 Similar Motions.** Our method exhibits limitations when handling parts with identical motion across states, as demonstrated in case 2 of Fig.5 where two drawers are pulled with the same distance. In such scenarios, the model tends to learn a single joint to fit both parts, failing to distinguish between the independently movable parts. As discussed in Sec.C, expanding ArtGS to incorporate additional states would provide richer motion information, potentially enabling better part separation.
>
> > Canonical Gaussian Evolution
>
> We visualize the evolution of canonical Gaussians in Fig.6, showing both their part assignments and centers. Our initialization strategy begins with dense static Gaussians and sparse dynamic Gaussians. As training progresses, the Gaussians undergo densification while simultaneously refining their part centers and assignments. These visualization results demonstrate the effectiveness of ArtGS.
>
> Note: Due to the addition of new visualizations, the interpolation results previously referenced as Fig. 6 in the latest revision are now shown in Fig. 8.
>
> We welcome further discussion regarding any remaining concerns.

---

### Meta-Review · Area_Chair_onJK · 2024-12-19

**Metareview:**

This paper introduces a method for part-aware articulated object reconstruction using 3D Gaussian splatting, with coarse-to-fine Gaussian initialization and self-guided motion priors. The reviewers agree on the strengths of the method, including its novel approach of part modeling, and integration of articulation learning. Extensive experiments validate its effectiveness and robustness. The coarse-to-fine initialization also improves its reconstruction quality.  During the discussion period, concerns from the reviewers were resolved, such as more qualitative results. However, concerns still remain about limited intermediate state results, writing clarity in the methods section, and broader generalization to diverse objects. I'd recommend the authors to address them in the revision.

**Additional Comments On Reviewer Discussion:**

There were several active rounds of discussion addressing concerns, such as clarifying the contributions, providing more visual results, improving the writing of the methods section, and expanding the discussion of the approach/failure cases. These concerns were partially resolved and acknowledged by the reviewers. The authors have committed to addressing all comments in the revision.

---

### Decision · Program_Chairs · 2025-01-22

Accept (Poster)